# Size uniformity of animal cells is actively maintained by a p38 MAPK-dependent regulation of G1-length

Shixuan Liu[1,2†], Miriam Bracha Ginzberg[1†], Nish Patel[1], Marc Hild[3], Bosco Leung[1], Zhengda Li[4], Yen-Chi Chen[5], Nancy Chang[1], Yuan Wang[3], Ceryl Tan[1,2], Shulamit Diena[1,2], William Trimble[1], Larry Wasserman[6], Jeremy L Jenkins[3], Marc W Kirschner[7]*, Ran Kafri[1,2]*

[1]Cell Biology Program, The Hospital for Sick Children, Toronto, Canada; [2]Department of Molecular Genetics, University of Toronto, Toronto, Canada; [3]Novartis Institutes for BioMedical Research, Cambridge, United States; [4]Department of Computational Medicine and Bioinformatics, University of Michigan, Ann Arbor, United States; [5]Department of Statistics, University of Washington, Seattle, United States; [6]Department of Statistics, Carnegie Mellon University, Pittsburgh, United States; [7]Department of Systems Biology, Harvard Medical School, Boston, United States

**Abstract** Animal cells within a tissue typically display a striking regularity in their size. To date, the molecular mechanisms that control this uniformity are still unknown. We have previously shown that size uniformity in animal cells is promoted, in part, by size-dependent regulation of G1 length. To identify the molecular mechanisms underlying this process, we performed a large-scale small molecule screen and found that the p38 MAPK pathway is involved in coordinating cell size and cell cycle progression. Small cells display higher p38 activity and spend more time in G1 than larger cells. Inhibition of p38 MAPK leads to loss of the compensatory G1 length extension in small cells, resulting in faster proliferation, smaller cell size and increased size heterogeneity. We propose a model wherein the p38 pathway responds to changes in cell size and regulates G1 exit accordingly, to increase cell size uniformity.

DOI: https://doi.org/10.7554/eLife.26947.001

*For correspondence:
marc@hms.harvard.edu (MWK);
ran.kafri@sickkids.ca (RK)

†These authors contributed equally to this work

## Introduction

Animal cells within a tissue often display a striking uniformity in cell size (*Ginzberg et al., 2015*; *Lloyd, 2013*). What are the mechanisms that ensure a single common target size for the numerous individual cells within a tissue? In other words, how is size variability suppressed? To date, most attempts to address this question have focused on the so-called cell size checkpoint – a mode of regulation that inhibits cell cycle progression for cells that are smaller than a target size (*Wood and Nurse, 2015*; *Neufeld and Edgar, 1998*).

In 1965, Killander and Zetterberg showed that populations of cultured mouse fibroblasts that are born small compensate with longer periods of growth in the G1 phase of cell cycle (*Killander and Zetterberg, 1965*). This suggests a model whereby size checkpoints selectively delay G1 progression of small cells, resulting in a negative correlation of birth size and G1 length. Following Zetterberg's early studies, additional evidence for size checkpoints in animal cells have been published by several other groups (*Shields et al., 1978*; *Darzynkiewicz et al., 1979*; *Gao and Raff, 1997*; *Dolznig et al., 2004*; *Tzur et al., 2009*; *Kafri et al., 2013*). Nevertheless, there remained skepticism

**eLife digest** Animal cells come in many different sizes. In humans, for example, egg cells are thousands of times larger than sperm cells. Yet cells of any given type are often strikingly similar in size. The cells that line the surface of organs including the skin and kidneys are especially uniform; in fact a loss of size uniformity in certain tumors is a sign of malignancy. What kind of regulation could enable separate cells within a tissue to have the same size?

One possibility is that each type of cell is programmed with a specific target size, and that a cell can sense if it strays from its target and compensate with longer or shorter periods of growth. Animal cells sensing their own size was first reported in the 1960s, and since then much research in this area has focused on "cell size checkpoints". These mechanisms stop cells that are too small from progressing through the series of events that allow one cell to divide in two, which is known as the cell cycle. Supporting the existence of size checkpoints, studies in yeast have repeatedly shown that cells that start off smaller tend to grow for longer during stages of cell cycle named G1 and G2.

Several researchers have proposed different mechanisms to explain how information about a cell's starting size influences the length of its cell cycle to result in the negative correlation between these two factors. However, as yet no one had managed to find a way to break this negative correlation, which would greatly help scientists to confirm the actual mechanisms that cells use to sense their size.

To address this, Liu, Ginzberg et al. first looked for chemicals that, when added to human cells, stopped cell size from being correlated with the time taken to complete a cell cycle. This search revealed that information about cell size is communicated to regulators of the cell cycle via a signaling pathway that involves an enzyme known as the p38 MAPK. Liu, Ginzberg et al. then showed that this specific pathway is activated in small but not large cells, where it slows the small cells' progress through the cell cycle. As expected, inhibiting the p38 enzyme also broke down the relationship between time spent in G1 and cell size, and led to the human cells growing to a range of different sizes.

These findings now pave the road to answering a fundamental question in cell biology: what is the elusive cell size sensor? Understanding how cells sense their size will open a window onto how quantitative information is programmed, sensed and communicated within living cells. These findings will shed also new light onto how cells specialize into cell types of different sizes, and what happens when cells lose the ability to sense or regulate their size in diseases like cancers.

DOI: https://doi.org/10.7554/eLife.26947.002

regarding the existence of a coordination between cell size and cell cycle in animal cells (*Lloyd, 2013*). This skepticism was largely, but not exclusively, owed to studies on Schwann cells which showed that growth and division can be influenced separately by different types of extracellular factors (*Conlon and Raff, 2003*; *Rathmell et al., 2000*). For a more comprehensive discussion of these disputes, see *Lloyd (2013)*.

A more substantial body of research on cell size checkpoints is found in studies on yeast (*Rupes, 2002*). In 1977, evidence supporting the existence of size checkpoints were demonstrated in both budding and fission yeast (*Fantes and Nurse, 1977*; *Johnston et al., 1977*). One point of difference between the two yeast species is that, while budding yeast compensates for smaller birth size by extending the duration of G1, fission yeast predominately (but not exclusively) employs a compensatory lengthening of G2. Nevertheless, the control mechanisms are similar: size uniformity is promoted by a mechanism that establishes a negative correlation between cell size and the duration of cell cycle. In fact, similar size-dependent regulation of cell cycle duration has also been discovered in bacteria (*Willis and Huang, 2017*) and plants (*Serrano-Mislata et al., 2015*).

How is cell size sensed and communicated to the cell cycle machinery? Two general class of models have been proposed: a geometric model and a titration model (*Wood and Nurse, 2015*). The geometric mechanism is, perhaps, best exemplified by a series of studies relating size of fission yeast to the localized activity of two proteins, Pom1 and Cdr2 (*Martin and Berthelot-Grosjean, 2009*; *Moseley et al., 2009*). In fission yeast *S. pombe*, Pom1 forms polar cortical gradients along the cell, peaking at the cell tips, while Cdr2 is localized to a medial band of cortical nodes. While still

controversial (*Wood and Nurse, 2013*), the suggestion is that inhibitory signals from the pole-localizing Pom1 declines as cell grow longer, allowing activation of Cdr2 and entry into mitosis. Several alternate geometric models of size sensing have been proposed and are reviewed in *Wood and Nurse (2015*). In addition to the geometric models, several titration models have been proposed in budding yeast *S. cerevisiae.* These models have focused on the G1 cyclin Cln3 and cell cycle inhibitor Whi5 as genetic perturbations of either genes significantly affects cell size (*de Bruin et al., 2004*; *Costanzo et al., 2004*). During G1, it was suggested that level of G1/S promoters such as Cln3 increase, while concentration of cell cycle inhibitors such as Whi5 decease due to dilution by growth of cell volume (*Wang et al., 2009*; *Schmoller and Skotheim, 2015*). These effects jointly determine the timing of S phase entry and cell size.

Despite the various models of mechanisms that correlate cell size and G1 length, a perturbation that breaks this correlation has not yet been reported. In the present study, we relied on a chemical screen which identified that in animal cells, inhibition of the p38 MAPK pathway results in loss of the coordination between size and G1 length. The mammalian p38 MAPK pathway participates in numerous biological processes, including the regulation of cell cycle checkpoints. In response to DNA damage or oxidative stress, p38 is activated and induces a cell cycle arrest (*Thornton and Rincon, 2009*; *Ambrosino and Nebreda, 2001*). Hyperosmotic conditions that shrink cell volume also strongly activate p38 (*Han et al., 1994*; *Moriguchi et al., 1996*; *New and Han, 1998*). Furthermore, upregulation of the p38 MAPK pathway can lead to increased cell size (*Clerk et al., 1998*; *Kudoh et al., 1998*; *Molnár et al., 1997*; *López-Avilés et al., 2005*; *Cully et al., 2010*). These observations raise the possibility that p38 may function to regulate cell size checkpoints in animal cells.

## Results

### A small molecule screen designed to perturb the coordination of cell size and cell cycle stage

To identify the molecular pathways linking cell size and cell cycle progression, we searched for perturbations that disrupt this link. We screened two compound libraries, known as the Novartis MOA (Mechanism-of-Action) box and Kinome box, which together included over 3000 compounds. The MOA Box contains an annotated list of compounds that are dynamically managed and curated to maximize coverage of targets, pathways, and bioactivity space (*Figure 1—figure supplement 1*). The MoA library was specifically designed to facilitate biological discovery by screening and profiling experiments. The Kinome Box is a library containing a wide range of kinase inhibitors that were selected based on their efficiency and specificity (primary targets IC50 <1 µM,<25 total targets). Uniquely, all compounds included in our screen are ones that are thoroughly annotated not only for their primary targets but also the lower affinity targets (off-targets). The advantage of this is that phenotypes associated with compound treatments can be statistically associated with particular signaling pathways despite the off-target effects that are inevitably associated with screened chemicals (see Materials and methods - Analysis of the compound screen).

To initiate the screen, unsynchronized HeLa cells were treated with compounds for 24 hr. After the compound treatments, we used a four-color labeling strategy to measure both cell size and cell cycle stage of each individual cell in each of the screened conditions. A combination of three labels (DAPI and the two FUCCI cell cycle reporters, mAG-hGem and mKO2-hCdt1) were used to report cell cycle stage. While mAG-hGem and mKO2-hCdt1 report progression through G1 (*Sakaue-Sawano et al., 2008*), DAPI labels progression through S phase. To measure cell size, we stained the cells with Alexa Fluor conjugated succinimidyl ester (SE) as a fourth color label. SE is a protein dye that accurately quantifies the total macromolecular protein mass in single cells and correlates with a cell's dry mass (*Kafri et al., 2013*).

Following compound treatments and the four-color labeling procedure, samples were imaged with fluorescence microscopy and images were processed with an automated image-processing pipeline that we developed to quantify fluorescence intensity per single cell (*Figure 1A*). With this screen assay, each drug treatment yielded a multi-dimensional joint distribution of cell size and cell cycle stage (*Figure 1B*). To simplify the analysis, we used this data to classify cells from each of the screened conditions into discrete cell cycle stages (*Figure 1—figure supplement 2*). In the end, this

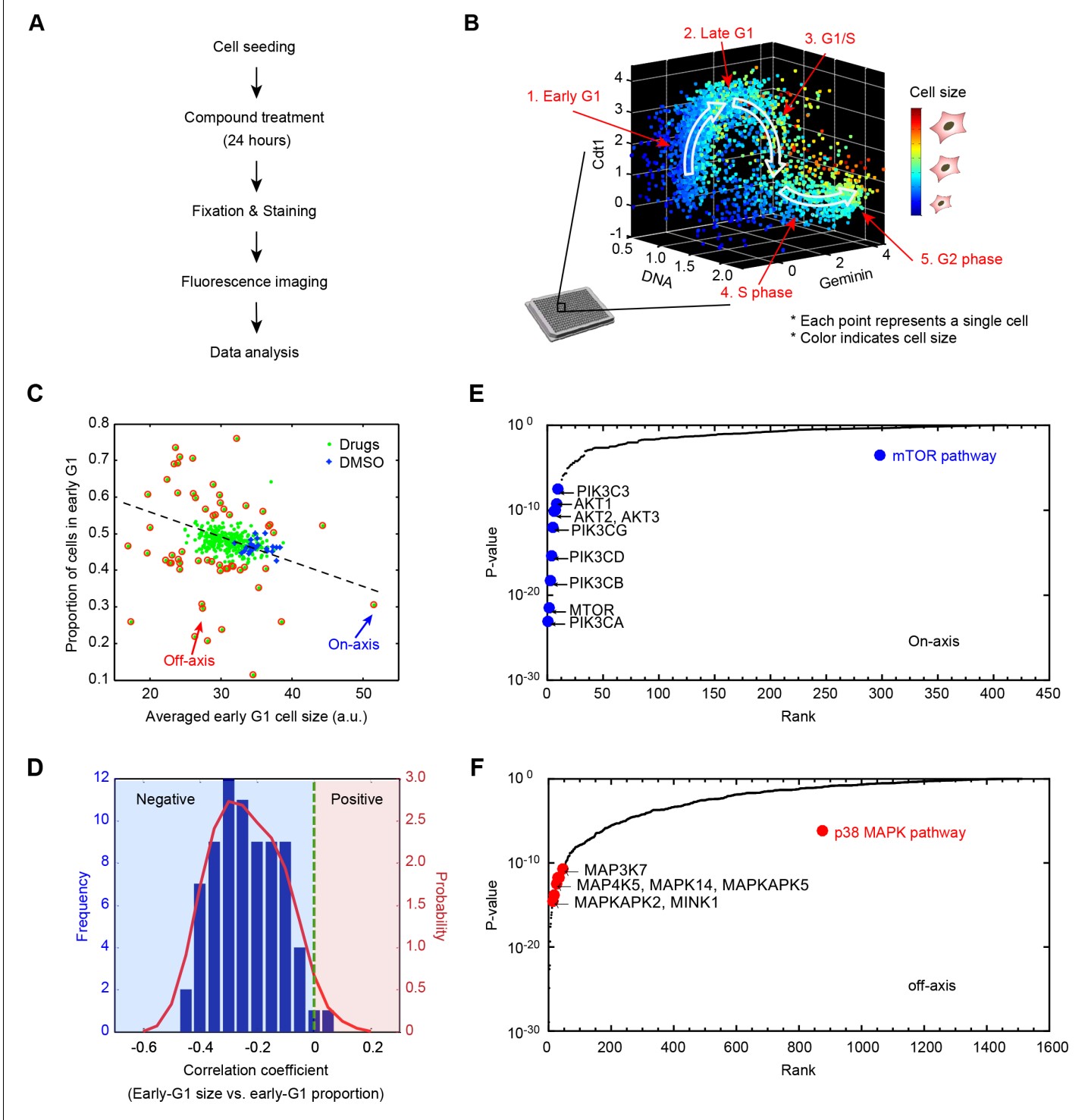

**Figure 1.** Results from a small molecule screen implicate the p38 MAPK pathway in the coordination of cell size and progression through G1. (A) A flowchart of the chemical screen demonstrating the major steps. (B) Raw data of cell size and cell cycle stage measured from a single control well in the screen. The scatter plot represents single-cell measurements of cell size and three markers of cell cycle stage; Cdt1 (mKO2-hCdt1), Geminin (mAG-hGem) and DNA (DAPI). Every point in the plot represents measurements on one single cell. Cell size is represented by a color scheme depicted by the colorbar on the right. White arrows are added to guide the reader along the cell cycle trajectory. (C) Average size of early G1 cells is negatively correlated with the fraction of cells in early G1. The scatterplot displays the result from one example 384-well plate. Each point on the plot corresponds to one particular screened condition (screened compound or control), and represents the average size of early G1 cells in that condition versus the proportion of cells in G1. Red circles highlight the conditions that significantly affect the size of early G1 cells and/or the proportion of cells in G1. The

*Figure 1 continued on next page*

*Figure 1 continued*

arrows designate examples of on-axis and off-axis compounds (also see *Figure 1—figure supplement 3*). (D) Distribution of correlation coefficients between average size of early G1 cells and the fraction of cells in G1, calculated for all screened plates (as described in Materials and methods - Analysis of the compound screen), demonstrating that the two variables are significantly negatively correlated (p<$10^{-16}$). (E, F) Ranked p-values from the target enrichment analysis of on-axis and off-axis compounds, respectively (Fisher's exact test). Components of the mTOR pathway and p38 MAPK pathway, which are highlighted, are among the top-ranked hits of on-axis and off-axis phenotypes, respectively. The Matlab script used to perform the target enrichment analysis is presented in *Figure 1—source code 1*.

DOI: https://doi.org/10.7554/eLife.26947.003

The following source data, source code and figure supplements are available for figure 1:

**Source data 1.** The screen metadata used to identify on-axis and off-axis outliers.
DOI: https://doi.org/10.7554/eLife.26947.009
**Source data 2.** The analysis script to visualize on-axis and off-axis outliers using *Figure 1—source data 1*.
DOI: https://doi.org/10.7554/eLife.26947.010
**Source code 1.** The Matlab script used to perform the target enrichment analysis.
DOI: https://doi.org/10.7554/eLife.26947.011
**Figure supplement 1.** Percentage of the target class coverage of the MOA Box compounds.
DOI: https://doi.org/10.7554/eLife.26947.004
**Figure supplement 2.** Cells from each well were partitioned, according to the three cell cycle indicators (DNA, Geminin, Cdt1), into discrete cell cycle stages.
DOI: https://doi.org/10.7554/eLife.26947.005
**Figure supplement 3.** Identification of on-axis and off-axis outliers.
DOI: https://doi.org/10.7554/eLife.26947.006
**Figure supplement 4.** Ranked p-values from the enrichment analysis of compounds that increase cell size variability (by Fisher's exact test).
DOI: https://doi.org/10.7554/eLife.26947.007
**Figure supplement 5.** The average z-score of cell size variability corresponding to each of the different target proteins.
DOI: https://doi.org/10.7554/eLife.26947.008

procedure resulted with detailed data on how each of the screened compounds influences cell count, average cell size, and size variability in each of the separate cell cycle stages.

## Screen hits are classified into *on-axis* and *off-axis* compounds

In an unsynchronized population, the proportion of cells in a particular cell cycle stage reflects the duration of that stage relative to the entire cell cycle (*Kafri et al., 2013*). Plotting the proportion of cells in early G1, from each of the screened conditions, against the average size of early G1 cells, revealed a negative correlation between the two quantities (*Figure 1C and D*). Chemical treatments that reduced the size of early G1 cells typically resulted in larger proportions of cells in G1, suggesting that the duration of G1 depends on a cell's initial size. To confirm this possibility, we used time-lapse microscopy to track nuclear size of live cells, which reaffirmed that cells born with smaller nucleus spend longer times in G1 (Figure 3G). In an accompanying manuscript authored by Ginzberg et al. (*Ginzberg et al., 2017*), we demonstrate that nuclear size is a faithful proxy of cell size. Altogether, these results support the hypothesis that G1 length is regulated in a size-dependent manner to promote uniformity among cells.

To investigate the pathways that link cell size and G1 length, we assembled the list of compounds that altered average cell size, the fraction of cells in G1, or both (i.e. outliers on the plot in *Figure 1C* and *Figure 1—figure supplement 3*, see Materials and methods - Analysis of the compound screen). We then sorted these 'hits' into two different categories. The first, which we call *on-axis* compounds, consisted of compounds that perturbed both cell size and the proportion of cells in G1, but did so in a manner that maintained the coordination between the two. In other words, on-axis compounds produced reciprocally correlated influences on cell size and the fraction of cells in G1. The second category of hits, which we call *off-axis* compounds, are compounds that disproportionally affected either size or cell cycle progression. Off-axis compounds resulted in paired values of cell size and G1 fraction that lie off-axis to the trend defined by the majority of screened conditions in *Figure 1C*. We reasoned that off-axis compounds may function by perturbing mechanisms that coordinate cell size with G1 length.

## Analysis of off-axis compounds suggests that p38 plays a role in the coordination cell size and G1 length

To identify the pathways associated with either the off-axis or on-axis phenotypes, we used the compound annotations to perform target-enrichment analysis with Fisher's exact test (see Materials and methods -Analysis of the compound screen). Specifically, we sought proteins and pathways that are significantly overrepresented in the combined target list of either the on-axis or off-axis compounds. This analysis avoids the risk of formulating a hypothesis based on a single compound. Interestingly, we found that the proteins most frequently targeted by the on-axis compounds are PI3K, Akt and mTOR (*Figure 1E*). This suggests that while the PI3K/Akt/mTOR pathway regulates cell size, this pathway is not involved in the coordination of cell size and the length of G1.

By contrast, the MAP Kinases are highly enriched among the targets of off-axis compounds (*Figure 1F*). Notably, multiple components of the p38 MAPK pathway, including p38α(MAPK14), MK2 (MAPKAPK2, a direct substrate of p38) and TAK1 (MAP3K7, an essential upstream activator of p38) are ranked in the top 1% of significant targets (p-value<0.01). Over 35 compounds that target p38 pathway components, highlighted in *Figure 1F*, consistently displayed an off-axis phenotype. This suggests that the p38 pathway is involved in the coordination of G1 length with cell size.

As an additional test for pathways that control cell size, we ranked compounds based on their influence on cell size variability (*Figure 1—figure supplements 4* and *5*). Using a similar target-enrichment approach, we generated a list of candidate proteins whose inhibition is associated with increased variation of cell size. This analysis showed that compounds that target p38 pathway components are associated with increased cell size variability. Notably, MK2, a downstream substrate of p38 was the highest-ranking target in this list. These results hint that p38 pathway may coordinate cell size with progression through G1 in a manner that promotes cell size uniformity.

## Inhibition of p38 disturbs the coordination of cell size and G1 length

To test whether the p38 MAPK pathway coordinates G1 length with cell size, we developed a robust assay to quantify this coordination in non-cancerous RPE1 cells (*Figure 2A*). The mTOR inhibitor, rapamycin, is an on-axis compound that perturbs cell size while maintaining the coordination of size with G1 length. As demonstrated by live-cell imaging, treatment with 5 nM rapamycin slows cellular growth rates by almost half (*Figure 3—figure supplement 1*). However, as seen in *Figure 3C and D*, this reduced growth causes only a slight decrease in cell size as its effect is largely buffered by an increased cell cycle duration. This phenomenon is further characterized in *Ginzberg et al., 2017*.

To quantify the strength of coordination between cell size and G1 length, we treated unsynchronized populations with a series of concentrations of rapamycin (see Materials and methods –Compound treatment). As shown in *Figure 2A*, higher concentrations of rapamycin result in smaller G1 cells and longer durations of G1. This experimental procedure reveals a robust and reproducible negative correlation between cell size and G1 length (*Figure 2A*). We then used this assay to test if p38 is required for the negative correlation of cell size with G1 length. We examined a panel of specific inhibitors against MAPK pathways (see Materials and methods –Compound treatment) and tested whether any of these inhibitors break the negative correlation induced by the rapamycin concentration series. Our results show that, consistently, inhibitors of p38 disrupt the negative correlation of size and the proportion of cells in G1 (*Figure 2B–E*, *Figure 2—figure supplements 1*, *2* and *3*). This result was observed with multiple chemical inhibitors of p38 that are distinct in chemical structure (*Gaestel et al., 2007*), making it less likely that the observed phenotype is owed to off-target effects. These results strongly implicate the p38 pathway in the coordination of G1 length with cell size. By contrast, inhibitors of JNK or ERK/MEK, do not significantly weaken the correlation of size and G1 length. Instead, these inhibitors shift the correlation to larger or smaller values of cell size (*Figure 2D and E*).

To further examine the involvement of p38 in the coordination between size and G1 length, we used time-lapse microscopy to track live cells and directly correlate their G1 length and birth size. Consistent with the conclusions of the bulk measurements, inhibition of p38 results in a weakened correlation of nucleus size and G1 length on a single-cell level (*Figure 3H and J*, *Figure 3—figure supplement 2*). Also, as expected from perturbations of size checkpoints, inhibition of p38 increases the variability in cell size (*Figure 3—figure supplement 3A*). In direct contrast to the effects of p38 inhibition, treating cells with the mTOR inhibitor, rapamycin, seems to strengthen the correlation of

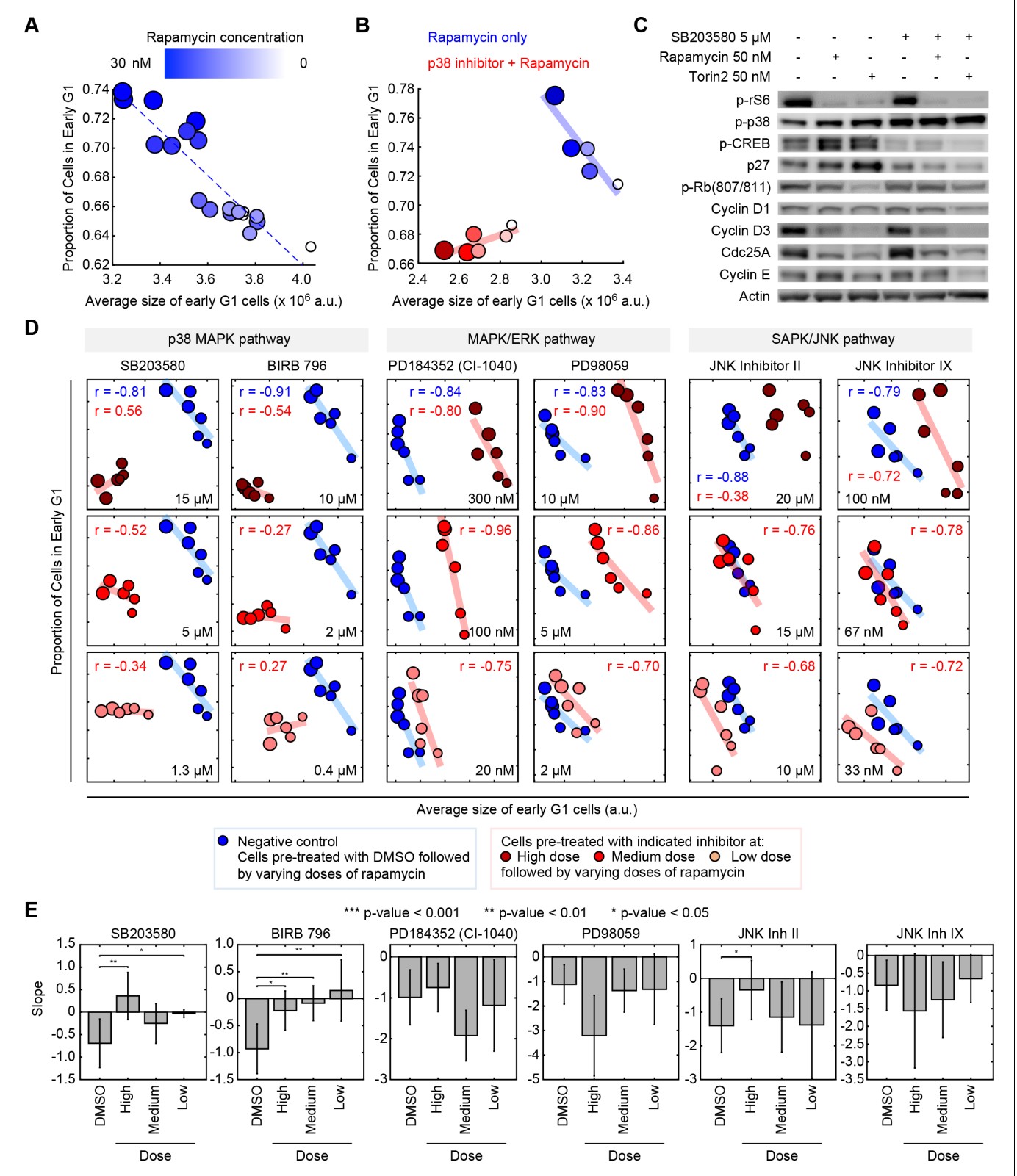

**Figure 2.** Pharmacological inhibition of the p38 MAPK pathway disrupts the coordination of cell size and G1 length. (**A**) Quantifying the coordination of cell size and G1 length. Samples of unsynchronized cells were treated with increasing concentrations of rapamycin (a rapamycin concentration series: 0, 0.03, 0.3, 3 and 30 nM) for a period of 24 hr, and then stained and imaged to quantify cell size and cell cycle stage on a single-cell basis. Each data point (circle) corresponds to a different concentration of rapamycin and shows the average size of early G1 cells and the proportion of cells in G1

*Figure 2 continued on next page*

*Figure 2 continued*

resulting from that treatment. Populations treated with higher concentrations of rapamycin had smaller cells and higher fractions of cells in G1, resulting in a robust negative correlation. Rapamycin concentrations are redundantly represented by both the size of the circles and their color, as shown in the colorbar. The small white circles represent control populations that were treated with DMSO, rather than rapamycin. Calculation of the average size and the proportion of G1 cells, in each of the represented samples, was performed by classifying single cells into cell cycle stage as depicted in *Figure 1B*. Each data point was measured from an unsynchronized population with a minimum of 7000 cells. Additional details on the experiment and analysis is provided in the Materials and methods section. (B) The experiment described in panel A is repeated with (red) or without (blue) a chemical inhibitor of p38 (SB203580, 5 µM). The negative correlation between the size of early G1 cells and the proportion of cells in G1 is apparent in populations not treated with SB203580 (blue) but not in the populations that are treated with SB203580. The blue and red trend lines represent linear regressions. (C) Western-blots of whole cell lysates from populations that were treated with different combinations of SB203580, rapamycin and Torin-2. The experimental procedure used here are the same as those used to generate the data shown in panel A and B. The increased levels of phopho-p38 in the population that is treated with SB203580 (a p38 inhibitor) should not be interpreted as a lack of efficacy of SB203580. Rather, these higher levels of phopho-p38 are explained by a negative feedback in the p38 pathway (*Arthur and Ley, 2013*), and the fact that while p38 inhibitors prevent p-p38 from phosphorylating its downstream substrates, these inhibitors do not block phosphorylation of p38 itself by upstream regulators (*Kumar et al., 1999*). (D) Inhibition of the p38 MAPK pathway, but not the MAPK/ERK or SAPK/JNK pathways, disrupts the correlation between the average size of early G1 cells and the proportion of cells in G1. Results were obtained with the same assay used to create panel A and B. Larger circle size indicates higher rapamycin concentration. The rapamycin concentration series includes: 0, 0.03, 0.1, 0.3, 3 and 30 nM. The results shown here are representative of three independent experiments. (E) Fitted slopes corresponding to the trends shown in *Figure 2D*. Error bars represent 90% confidence intervals. For each compound treatment, its fitted slope is compared with the slope of the control (DMSO) from the same experiment. Significance was calculated with one-tailed Student's *t*-test ($H_0$: slope$_{drug}$ <= slope$_{control}$). The meta data and source code used for this analysis and visualization of results is presented in *Figure 2—source data 1*.

DOI: https://doi.org/10.7554/eLife.26947.012

The following source data and figure supplements are available for figure 2:

**Source data 1.** Measurements of cell size and cell cycle stages from the chemical inhibitor experiments as shown in *Figure 2D*, *Figure 2—figure supplements 2* and *3*.
DOI: https://doi.org/10.7554/eLife.26947.017

**Figure supplement 1.** Western-blot of cell lysates from conditions shown in *Figure 2C* confirms the chemical inhibitors are efficient towards inhibiting corresponding MAPKs pathway.
DOI: https://doi.org/10.7554/eLife.26947.013

**Figure supplement 2.** The negative correlation between cell size and proportion of cells in early G1 is perturbed or weakened under p38 inhibition.
DOI: https://doi.org/10.7554/eLife.26947.014

**Figure supplement 3.** Inhibitors of p38 display dose-dependent influence in the coordination of cell size and G1 length.
DOI: https://doi.org/10.7554/eLife.26947.015

**Figure supplement 4.** Quantification of p-p38, p-CREB and p27 in conditions shown *Figure 2C*.
DOI: https://doi.org/10.7554/eLife.26947.016

size and G1 length (*Figure 3I*). One possible reason for this is that, by shrinking cells, rapamycin subjects a greater proportion of cells to the control of a size checkpoint.

In addition to weakening the correlation of G1 length and cell size, p38 inhibitors also cause a reduction in average cell size (*Figure 2B and D*). To understand this cell shrinkage, we performed two sets of complementary measurements, both aimed at examining the influence of p38 inhibition on cell growth and cell cycle progression. In the first set of measurements, we used time-lapse microscopy to collect proliferation curves and growth curves of single cells throughout their entire cell cycle. Imaged cells expressed mAG-hGem, a cell cycle reporter (*Sakaue-Sawano et al., 2008*), enabling accurate detection of the G1/S transition (see Materials and methods - Automated lineage tracking). In the second set of measurements, we performed time-course experiments to periodically collect samples of unsynchronized proliferating cells. For each collected sample, we measured the total bulk protein mass and total cell count, which were then used to estimate the average rate of cell growth and the average rate of cell division (see Materials and methods - Estimation of cell cycle durations and growth rate from bulk measurements).

As shown in *Figure 3*, p38 inhibition results in shorter cell cycles (*Figure 3B and D*, *Figure 3—figure supplement 3B*), with a consequential increase in rates of cell division (*Figure 3A*). In addition, the results show that the shortened cell cycle length caused by p38 inhibitors is owed to a decrease in G1 length (*Figure 3E* and *Figure 3—figure supplement 3C*) with no significant changes in the lengths of S or G2 phases (*Figure 3F*, *Figure 3—figure supplement 3D E*). At G1/S transition, cells subject to p38 inhibition have smaller nucleus size as compared to control (*Figure 3C*). This decrease

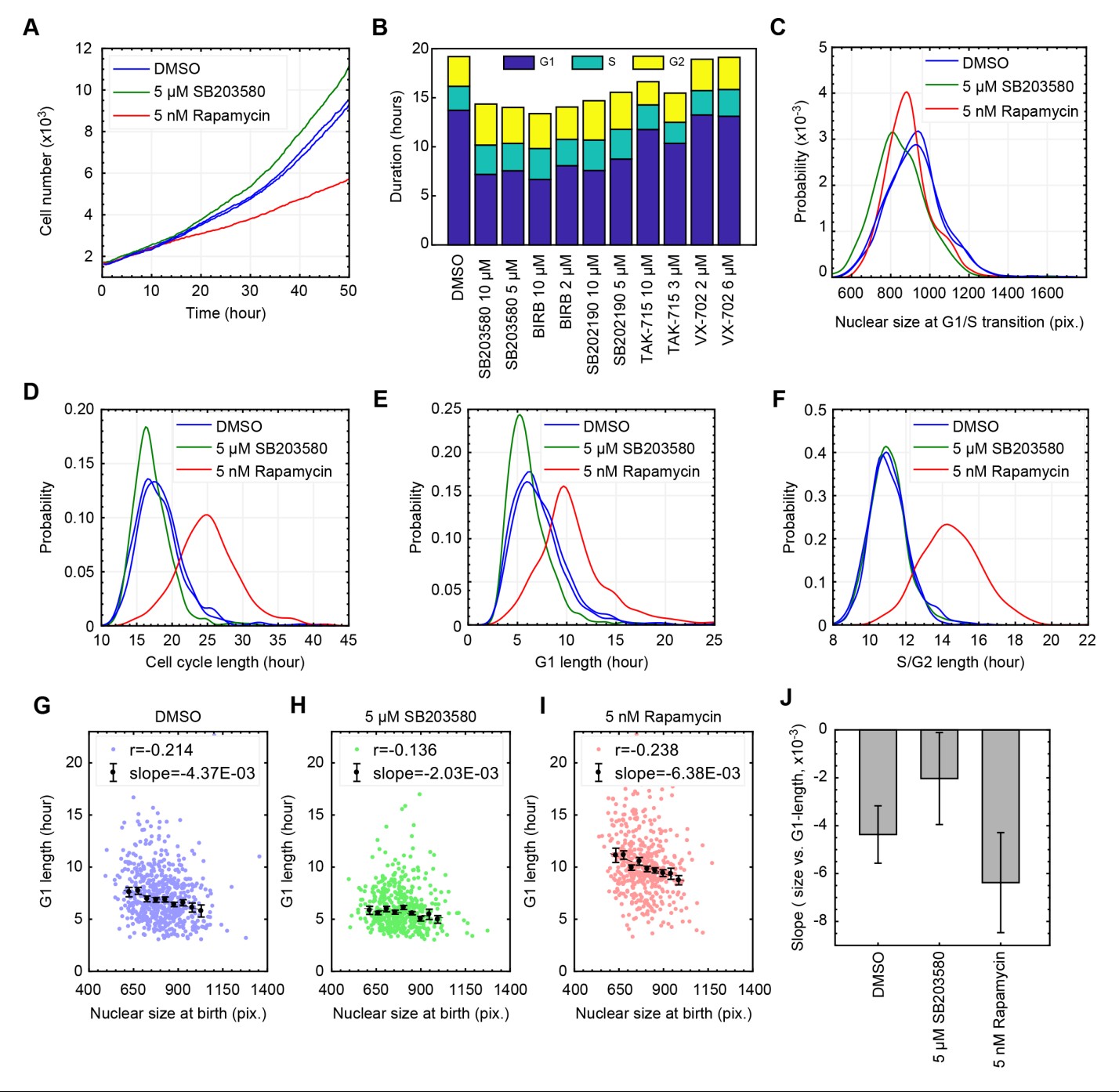

**Figure 3.** Inhibition of p38 weakens the coordination of cell size and G1 length at a single-cell level. (**A**) Live cells subject to p38 inhibition (SB203580) or to mTORC1 inhibition (rapamycin) were followed with time-lapse microscopy to monitor proliferation over a period of 50 hr. mTOR inhibition significantly slowed rates of proliferation, while p38 inhibition increased rates of proliferation. (**B**) As an alternative method to assay cell cycle lengths, populations of cells were treated with p38 inhibitors and samples were fixed every 20 hr over a period of 3 days (see Materials and methods -Estimation of cell proliferation durations and growth rate from bulk measurements). Proportion of cells in the different cell cycle stages, in each of the collected samples, were calculated based on the cell cycle indicators depicted in *Figure 1B*. Consistently, p38 inhibitors accelerate proliferation by shortening the duration of G1 but not the durations of S or G2 (also see *Figure 3—figure supplement 3*). The meta data and source code used for this cell cycle analysis is presented in *Figure 3—source data 1*. (**C–F**) Live cells were imaged by time-lapse microscopy for a period of 50 hr to obtain growth trajectories of single cells over the course of their entire cell cycle. Computer generated image processing and cell tracking were performed, as described in Materials and methods - Automated lineage tracking and analysis, to obtain single cell growth curves. Nuclear size was used as a proxy of cell size, as has been validated in *Ginzberg et al., 2017*. Cells that were successfully tracked throughout their entire cell cycle were collected to

*Figure 3 continued on next page*

*Figure 3 continued*

calculate the cell cycle durations and cell size dynamics. (**G–I**) Scatterplots displaying relationship between nuclear size at birth and G1 duration for individual cells that are subject to chemical inhibition of p38 (**H**), chemical inhibition of mTOR (**I**), and a control population treated with DMSO (**G**). Every single point corresponds to the birth size and G1 length of a single live cell that was followed by time-lapse microscopy. Also shown are means and errorbars (SEM) of average G1 length calculated for different cell size bins. The dashed line shows the result of linear regression with the binned data. (**J**) Slopes obtained by the linear regression shown in (**G–I**) Error bars indicate 95% confidence bounds. The results shown here are representative of two independent experiments. The single-cell tracking data from the live-cell imaging experiments and the source code for analysis and visualization of the results is presented in *Figure 3—source data 2*.

DOI: https://doi.org/10.7554/eLife.26947.018

The following source data and figure supplements are available for figure 3:

**Source data 1.** Estimation of cell cycle duration and growth rate from bulk measurements of fixed cell populations.

DOI: https://doi.org/10.7554/eLife.26947.022

**Source data 2.** Measurements of single-cell dynamics of cell size captured by live-cell imaging.

DOI: https://doi.org/10.7554/eLife.26947.023

**Figure supplement 1.** Cellular growth rate also negatively correlates with G1 or cell cycle duration.

DOI: https://doi.org/10.7554/eLife.26947.019

**Figure supplement 2.** Cell size at birth is negatively correlates with cell cycle duration. p38 inhibition, but not mTORC1 inhibition weakens this correlation.

DOI: https://doi.org/10.7554/eLife.26947.020

**Figure supplement 3.** Inhibition of p38 MAPK increases cell size variability (**A**), promote proliferation (**B**) by shortening G1 length (**C**) without significant effect in S/G2 duration (**D and E**) or cellular growth rate (**F**).

DOI: https://doi.org/10.7554/eLife.26947.021

in cell size is caused by a shorter growth duration (in G1) rather than a slower growth rate. This conclusion is independently demonstrated by each of the two experiments described above, both showing similar growth rates for p38-inhibited cells and control cells (*Figure 3—figure supplement 3F*, *Figure 3—figure supplement 1*). Altogether, this suggests that the p38 pathway mediates a cell size checkpoint at the G1 transition, selectively allowing large but not small cells to progress into S phase. Potentially, this model also explains why p38 inhibitors reduce cell size. By disrupting the cell size checkpoint, p38 inhibitors cause cells to progress through the G1/S transition with prematurely small size, causing an overall reduction in cell size.

## Genetic knockdown of p38 pathway components disturbs the coordination of cell size and G1 length

Mammalians have four isoforms of p38 MAPK (p38α, p38β, p38γ and p38δ). Using the same assay described in *Figure 2A*, we examined how knockdown of the p38 MAPKs affects the coordination of cell size and G1 length. Interestingly, using siRNA to knockdown p38γ or p38δ, significantly weakens or eliminates the negative correlation between cell size and proportion of cells in G1, while knockdown of p38α and p38β only partially weakens this correlation (*Figure 4A and B*, *Figure 4—figure supplements 1* and *2*). Faced with this result, one may wonder why chemical inhibitors of p38α had such a dramatic influence on a coordination that is mediated primarily by p38γ and p38δ? As shown in two comprehensive studies (*Davis et al., 2011*; *Karaman et al., 2008*), two of tested p38 inhibitors (SB203580 and BIRB-796) also target p38γ and/or p38δ (*Figure 4—source data 1*) at their working concentrations (*Figure 4—figure supplement 2*). Combined with the siRNA knockdown of the p38 isoforms, it suggests that the coordination between cell size and G1 progression is mainly mediated through p38γ and p38δ.

An additional difference between the chemical and genetic inhibition of p38 is a greater reduction in cell size that results from the chemical inhibition. A possible explanation for this difference is the lower specificity of chemical inhibitors as compared to genetic perturbations. Functional redundancy of the different p38 isoforms, for example, can partially compensate siRNA treatments but not pharmacological perturbations. Alternatively, it may be that transfection with siRNA activates nonspecific cellular responses, such as upregulation of innate immune pathways, that lead to an increase in size (*Snijder et al., 2012*; *Marques and Williams, 2005*; *Grumont et al., 2004*). Nonetheless, these results collectively demonstrate that both chemical and genetic inhibition of p38 activity result in loss of coordination between cell size and G1 length.

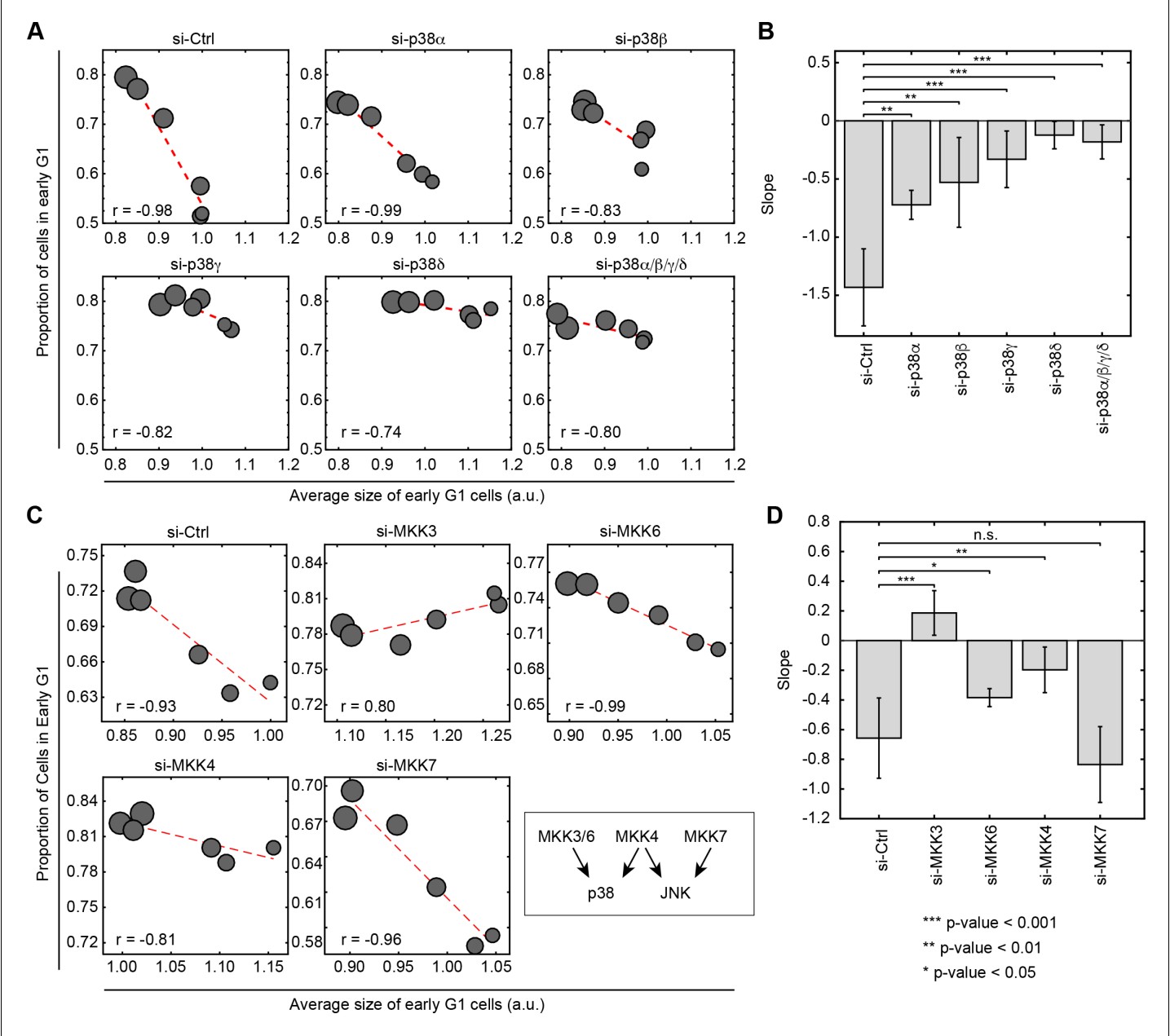

**Figure 4.** Knockdown of p38 pathway components disturbs the negative correlation between cell size and proportion of cells in G1. Cells were transfected with siRNA as indicated and subsequently assayed with a rapamycin concentration series (0, 0.03, 0.1, 0.3, 3 and 30 nM) as described in *Figure 2A* to assay the correlation of size and G1 length. Larger circle size indicates higher concentrations of rapamycin. (A) Knocking down p38α/β partially weakens the negative correlation between cell size and proportion of cells in G1, while knockdown of p38γ/δ drastically disturbs the correlation. (C) The negative correlation between cell size and proportion of cells in G1 is disturbed when cells are transfected with siRNA against MKK3/4/6 but not MKK7. Each data point in *Figure 4A and C* is measured on an unsynchronized population with a minimum of 3000 cells. The results shown in *Figure 4A and C* are representative of two and three independent experiments with duplicates or triplicates. (B, D) Fitted slopes of the trends shown in *Figure 4A and C*. Error bars indicate 90% confidence intervals. Analysis is performed with the same method as indicated in *Figure 2E*. The meta data and source code to analyze and visualize the genetic knock down results is presented in *Figure 4—source data 1*.
DOI: https://doi.org/10.7554/eLife.26947.024

The following source data and figure supplements are available for figure 4:

**Source data 1.** Binding activity (Kd's in nM) of the p38 inhibitors used in the study against each of the p38 isoforms.
DOI: https://doi.org/10.7554/eLife.26947.027
**Source data 2.** Measurements of cell size and cell cycle stage from the knockdown experiments as shown in *Figure 4*.
DOI: https://doi.org/10.7554/eLife.26947.028
*Figure 4 continued on next page*

*Figure 4 continued*

**Figure supplement 1.** Western-blot of cell lysates from conditions shown in **Figure 3** confirms efficiency of knockdown of MKKs (**A**) or p38 isoforms (**B**).
DOI: https://doi.org/10.7554/eLife.26947.025
**Figure supplement 2.** Cells are still cycling upon the knockdown treatments.
DOI: https://doi.org/10.7554/eLife.26947.026

To further test the relationship of p38 and the control of cell size, we used siRNA to perturb the activity of MKK3/6/4, which are upstream activators of the p38 pathway. As a control, we also knocked down MKK7, an upstream regulator of JNK. Consistent with the results of the chemical inhibitors, knocking down either MKK3, MKK6 or MKK4 disturbs the negative correlation between cell size and proportion of cells in G1 (**Figure 4C and D**, **Figure 4—figure supplement 1**). By contrast, knocking down MKK7 leaves the coordination of G1 and cell size intact. This result further supports a model whereby the coordination of cell size and cell cycle is specific to p38 and not the other MAPK pathways.

## p38 mediates the size-dependent G1 length extension by influencing G1/S regulators

The p38 MAPK pathway mediates cell cycle checkpoints by modulating multiple cell cycle regulators including cyclin D, cyclin E, p53 and CDC25 (**Thornton and Rincon, 2009**; **Lavoie et al., 1996**; **Mikule et al., 2007**; **Yee et al., 2004**). To test the mechanisms of the p38-mediated cell size checkpoint, we treated cells with inhibitors of mTOR, rapamycin and Torin-2, to induce a size-dependent lengthening of G1. Several cell cycle regulators were assayed including positive indicators of G1/S transition (Cyclin D, Cyclin E, p-Rb, Cdc25A) and negative regulators of G1 progression (p27$^{Kip1}$) (**Figure 2C**). Surprisingly, inhibition of p38 reversed the influence of mTOR inhibition on p27 (**Figure 2C**, **Figure 2—figure supplement 4**). Cells subject to mTOR inhibition display increased levels of p27, a CDK inhibitor that functions to prolong G1. By contrast, when these mTOR-inhibited cells are co-treated with inhibitors of p38, p27 levels are reduced. These results raise the intriguing possibility that the p38-dependent lengthening of G1 in small cells is mediated by the cell cycle inhibitor, p27. However, it is worth noting that, since p38 has long been implicated with the G1 arrest that follows environmental stress conditions, several different G1 regulators in addition to p27 have been previously identified as p38 effectors (**Thornton and Rincon, 2009**; **Lavoie et al., 1996**; **Mikule et al., 2007**; **Yee et al., 2004**). It is, therefore, not unlikely that the p38 mediated G1 lengthening is redundantly mediated by multiple effectors, making this connection an intriguing subject for future investiations.

## Cell-size-dependent activation of the p38 pathway

The model suggesting that p38 regulates a cell size checkpoint is contingent on the hypothesis that the activity of the p38 MAPK pathway is somehow dependent on cell size. **Figure 2C** shows that p38 is, indeed, activated when cell size is decreased by inhibition of mTOR. While this result is consistent with size-dependent p38 activation, it could also have an alternative explanation. Namely, it may be that p38 is directly activated by molecular signals resulting from mTOR inhibition, in a manner that is independent of the influence of mTOR on cell size. To test this possibility, we relied on the difference in the time scales associated with cell growth and the time scales associated with mTOR activity. As shown in **Figure 5A**, we designed a 'size-recovery experiment' in which cells were released from a 20 hr Torin-2-mediated inhibition of mTOR and allowed to recover in size. The prolonged mTOR inhibition reduced average cell size by over 15%. Note that cells subject to Torin-2 treatment were still cycling (**Figure 5—figure supplement 1**). When Torin-2 is washed out, mTOR activity is restored within 1 hr (**Figure 5B**), whereas recovery of cell size proceeds slowly, over a period of 14 hr. This experimental design provides an opportunity to observe cells that are smaller than their target size, but have long been relived from mTOR inhibition (**Figure 5B**). Western-blots of lysates from cells during the recovery phase show that the p38 MAPK remains upregulated long after mTOR has resumed normal levels of activity (**Figure 5B**) and that the dynamics of p38 activity mirror the dynamics of the recovery of cell size rather than the recovery of mTOR activity.

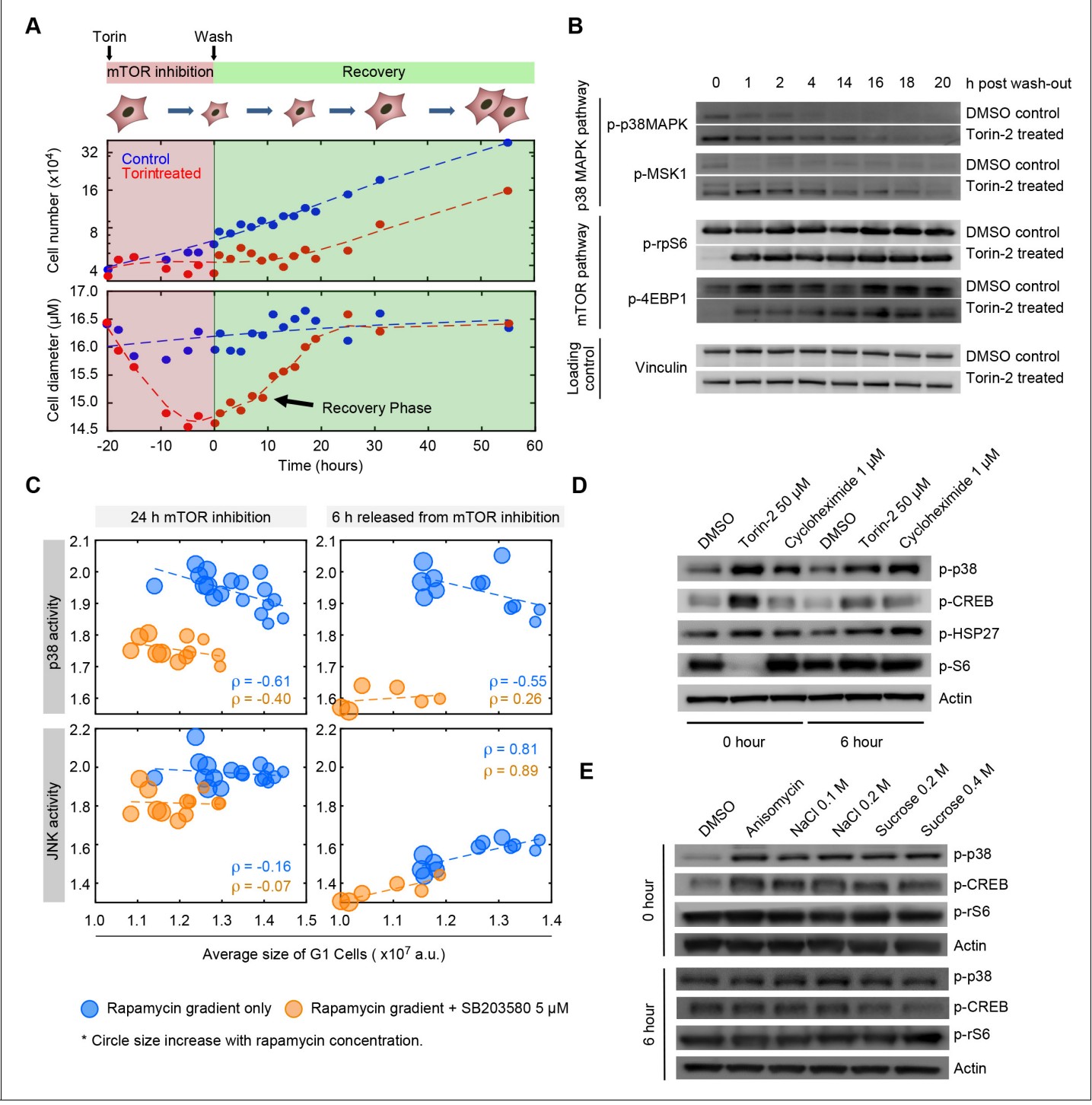

**Figure 5.** The p38 MAPK pathway is selectively upregulated in small cells. (A) Cells were treated with either 50 nM of Torin-2 or DMSO (control) for 20 hr, followed by drug wash-out and media replacement. Cells undergoing mTOR inhibition, on average, decrease in size and slow their proliferation rate. Following release from mTOR inhibition, cells grow but maintain a low proliferation rate until their normal size is reached. Cells resume a wild type rate of proliferation only when their size reaches the size of the untreated population. (B) Western blots of whole cell lysates collected at time points ranging from 0 to 20 hr post release from mTOR inhibition. Levels of mTOR pathway activity recover within 1 hr after Torin-2 wash-out. By contrast, activity of p38 remains upregulated in the Torin-treated cells compared with controls, and gradually fades away only as cells recover their wild-type size. (C) Cells simultaneously expressing reporters of both p38 MAPK and JNK were treated with a series of rapamycin concentrations, as in **Figure 2A**. Each data point (circle) corresponds to the average G1 cell size and the average level of MAPK activity (JNK and p38) that corresponds to a given concentration of rapamycin. As positive controls, we include populations that were co-treated with the p38 inhibitor, SB203580 (orange circles). Higher

*Figure 5 continued on next page*

*Figure 5 continued*

concentrations of rapamycin (bigger circle size) result in smaller cells with higher activity of p38 (top left panel). Unlike p38, activity of JNK was not upregulated in proportion to cell size (left bottom panel). Also shown are the correlations of MAPK activity (JNK and p38) and cell size at 6 hr post release from mTOR inhibition (right panels) (also see *Figure 5—figure supplement 2*). Each data point represents average values of cell size and MAPK activity of the G1 cells subpopulation from an unsynchronized population with a minimum of 3000 cells. Results shown here are representative of three independent experiments. The meta data and source code used to analyze and visualize the correlation between cell size and KTR readout is presented in *Figure 5—source data 1*. (D) Western-blots of whole cell lysates from samples collected at 0 or 6 hr post release from a 22 hr treatment with either 50 nM Torin-2, 1 µM cycloheximide or DMSO (control). (E) Western-blots of whole cell lysates from samples collected at 0 or 6 hr post release from a 30-min treatment with either 25 ng/mL anisomycin, hyperosmotic shocks (NaCl and Sucrose) or DMSO (control).
DOI: https://doi.org/10.7554/eLife.26947.029

The following source data and figure supplements are available for figure 5:

**Source data 1.** Measurements of cell size and p38 KTR as shown in *Figure 5C* and *Figure 5—figure supplement 4*.
DOI: https://doi.org/10.7554/eLife.26947.035
**Figure supplement 1.** Cells are still cycling upon Torin-2 treatment.
DOI: https://doi.org/10.7554/eLife.26947.030
**Figure supplement 2.** Representative images showing response of the p38 KTR to indicated treatments.
DOI: https://doi.org/10.7554/eLife.26947.031
**Figure supplement 3.** Immunofluorescence images of cells stained with a phospho-p38 antibody after indicated treatments.
DOI: https://doi.org/10.7554/eLife.26947.032
**Figure supplement 4.** p38 activity negatively correlates with cell size in G1 but not S or G2 cells.
DOI: https://doi.org/10.7554/eLife.26947.033
**Figure supplement 5.** Average cell size measured by Coulter counter after the cells were released from the indicated compound treatment.
DOI: https://doi.org/10.7554/eLife.26947.034

To further test whether p38 is selectively upregulated in small G1 cells, we employed a p38 Kinase Translocation Reporter (KTR) (*Regot et al., 2014*) that quantitatively reports p38 activation in live cells. This assay involves a fluorescent reporter that, once phosphorylated by p38 MAPK, shuttles from the nucleus to the cytoplasm. The ratio of cytoplasmic to nuclear (C/N) fluorescence represents a quantitative measure of p38 activation (see Materials and methods -Image processing and cell segmentation). In addition to the KTR, cells were also stained with DAPI to discriminate the G1, S and G2 subpopulations.

As expected, cells treated with a p38 inhibitor display a lower C/N ratio as compared to untreated cells (*Figure 5C*, *Figure 5—figure supplement 2*). Using a similar assay as described in *Figure 2A*, cells expressing the p38 KTR were treated with increasing concentrations of rapamycin for a period of 22 hr, resulting in a dose-dependent decrease in cell size. Consistent with our hypothesis, decreases in cell size lead to upregulated p38 activity (*Figure 5C*, *Figure 5—figure supplements 2,3*). To exclude the possibility that differences in KTR localization are a direct result of mTOR inhibition rather than the influence of mTOR inhibition on cell size, we also performed measurements on cells six hours after the mTOR inhibitor was washed out. Confirming our hypothesis, the negative correlation of cell size and p38 activity persists for at least 6 hr after cells are released from mTOR inhibition, long after normal mTOR activity is resumed. Interestingly, in cells released from mTOR inhibition, p38 activity correlates with cell size only in G1, but not in S or G2 (*Figure 5—figure supplement 4*). This result is consistent with a model whereby size-dependent p38 activity regulates the G1/S transition. As an additional control, we expressed dual color KTRs in the cells, reporting the activity of both p38 and JNK. These measurements showed that while p38 activity is negatively correlated with cell size, JNK activity is not (*Figure 5C*).

The sustained p38 activity, which proceeds hours after mTOR activity has resumed normal levels of activity, supports the hypothesis that it is the reduction in cell size and not inhibition of mTOR that activates p38. However, an alternative possibility is that the sustained p38 activity is due to intrinsically slow inactivation kinetics of the p38 pathway, in a manner that is independent of cell size. To test this possibility, we exposed cells to short treatment of anisomycin or hyperosmotic shock, both of which strongly activate p38 (*Figure 5E*) without interfering with cell mass. Note although hyperosmotic shock does reduce cell volume, it does not reduce cell mass. Cell mass is the product of anabolic processes, largely regulated by mTOR, that changes over time scales of hours. By contrast, cell volume is a labile phenotype that changes over short time scales and is regulated

by various ion channels and transporters (*Lang et al., 1998*). Further, as a complimentary approach, we exposed cells to cycloheximide, a protein-translation inhibitor that significantly reduces cell mass but does not inhibit mTOR (*Figure 5D*). In fact, as previously reported (*Bar-Peled and Sabatini, 2014*; *Sancak et al., 2008*), cycloheximide activates mTOR due to increased intracellular pools of amino acids.

*Figure 5D and E* show that sustained p38 activity, which persists after perturbations are removed, is observed only in conditions that decreases cell size. Specifically, 6 hr after cells had been released from the chemical treatments, increased p-p38 level was observed in cells that had been treated with Torin-2 or cycloheximide (*Figure 5D*, *Figure 5—figure supplement 5*), while p-p38 level of cells that had been treated with anisomycin or osmotic shock was indistinguishable from control (*Figure 5E*). These results suggest that the sustained activity of p38 accompanying the recovery of cell size is a result of reduction in cell size rather than mTOR inhibition or slow inactivation kinetics of p38 signaling.

## Inhibition of p38 represses recovery of cell size

As a final test for the hypothesis that p38 plays a role in the maintenance of cell size, we asked if p38 activity is required for cells that are recovering from perturbations of cell size. To test this, we repeated the size recovery experiment shown in *Figure 5A*, in the presence or absence of several different p38 inhibitors (*Figure 6A*). *Figure 5A* shows that it takes approximately 25 hr for cells to fully recover their size following release from Torin-2 inhibition. We suspected that p38 inhibitors may either delay this process or inhibit recovery altogether. To discriminate these two options, we performed cell size measurements not only at times that fall within the normal time scales of recovery (6 hr, 24 hr), but also at time points that significantly exceed this range (30 hr, 48 hr). Indeed, our results show that even at the late time point of 48 hr after Torin washout, cell size is significantly reduced. Anecdotally, while p38 inhibition prevents the recovery in cell size at the late time points, a partial recovery is observed at 6 hr post Torin washout. This partial recovery, at early time points, is not unexpected. When cells are subject to mTOR inhibition (Torin-2), their growth rate is significantly reduced. Once mTOR inhibition has ceased, growth rate leaps back to its normal high levels, causing an immediate increase in cell size that is apparent in the early time points (6 hr post Torin-2 washout). However, to fully recover their size, cells must also lengthen the duration of growth by invoking the cell size checkpoint in G1. As our results suggest, it is this process that becomes defective when p38 is inhibited, which manifests its influence on size only at the later time points.

As additional controls, we also used inhibitors against two other MAPKs, ERK and JNK. Consistent with our expectation, inhibition of JNK or ERK do not repress recovery in cell size (*Figure 6B*). Further, inhibition of p38 increased proliferation rates as compared to control (*Figure 6C*), supporting the hypothesis that p38 activation inhibits cell cycle progression of cells that are smaller than their appropriate size. Interestingly, the influence of p38 inhibition on the recovery in cell size persists long after the inhibitors are washed out (*Figure 6—figure supplement 1*). While Torin-2-treated cells begin to recover their size immediately after mTOR inhibition is relieved, cells that are co-treated with both Torin-2 and a p38 inhibitor display a marked delay in recovery (*Figure 6—figure supplement 1*). This delay may hint that the cell growth cycle depends on a commitment point that is cell-cycle-stage-dependent, analogous to the restriction point that regulates the cell division cycle (*Blagosklonny and Pardee, 2002*).

## Discussion

It has been over five decades since the first discovery of the connection between cell size and cell cycle in mammalian cells (*Killander and Zetterberg, 1965*). However, the molecular mechanism underlying this coordination between size and cell cycle remained elusive. In this study, we relied on a small molecule screen to identify that the p38 MAPK pathway is involved in the coordination cell cycle progression with cell size. At the heart of our methodological approach was an analysis that identified an axis of coordination relating cell size and cell cycle progression (*Figure 1C*). Namely, we showed that almost all screened conditions that decrease cell size also bring about an increase in the proportion of G1 cells. Based on this, we classified the screen hits into two categories: on-axis compounds and off-axis compounds. Compounds that perturb cell size or cell cycle progression but maintain a coordination between the two were called on-axis compounds. By contrast, compounds

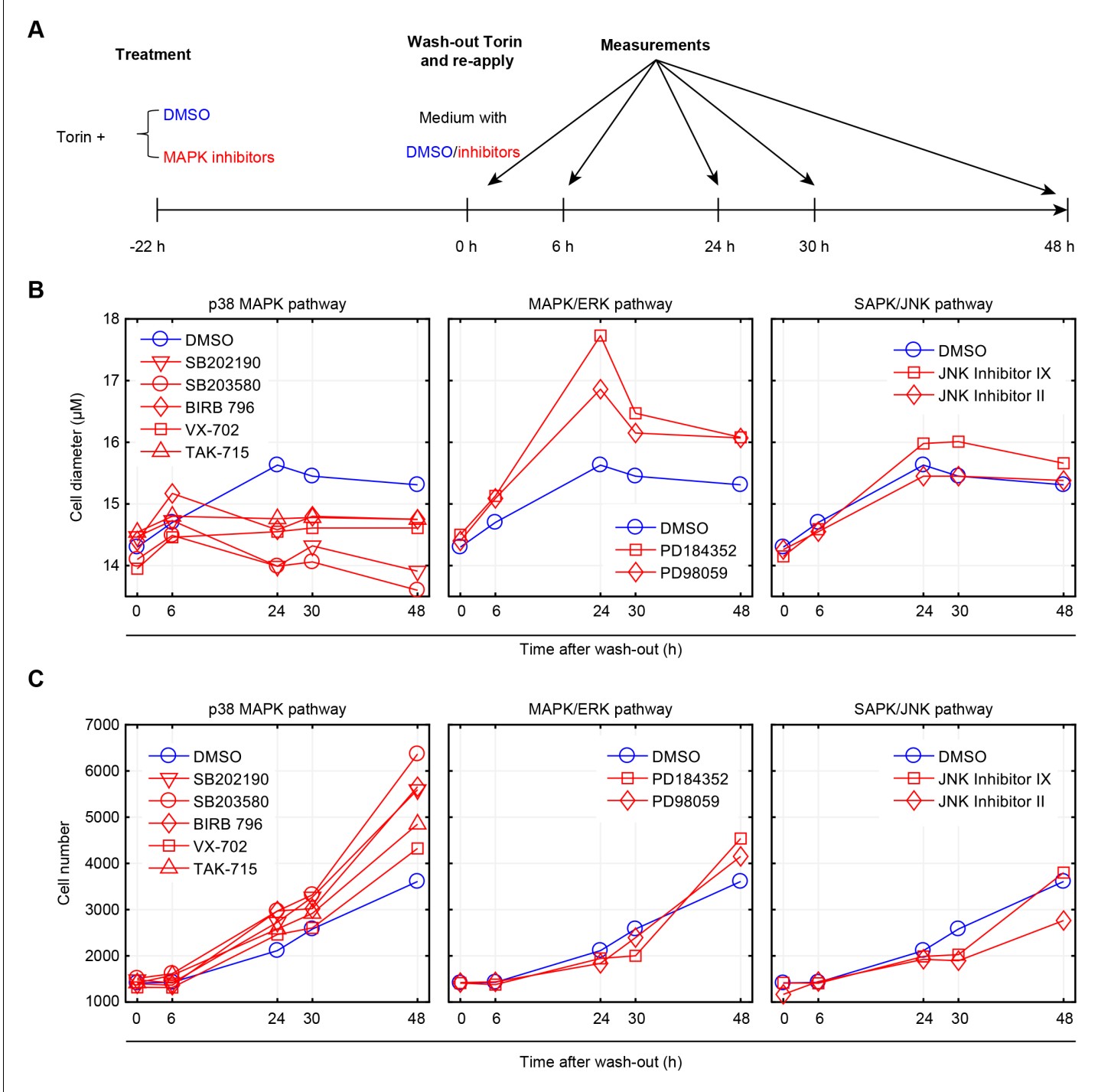

**Figure 6.** Inhibition of p38, but not ERK or JNK, represses recovery of size in cells released from mTOR inhibition. (**A**) Workflow of the experiment. Cells were treated with 50 nM Torin-2 with or without the indicated MAPK inhibitors for 22 hr, and then released from Torin-2 while still being subject to the indicated MAPK inhibitors (red) or DMSO (blue). MAPK inhibitors were administrated at a concentration consistent with the highest corresponding concentration used in *Figure 2* and *Figure 2—figure supplement 2*. At 0, 6, 24, 30 and 48 hr post release from Torin-2 treatment, samples were measured for both average cell size (**B**) and cell count (**C**). (**B**) Cells treated with DMSO (control) recovered in size within 24 hr and remained at a constant average size thereafter. Cells treated with p38 inhibitors, but not ERK or JNK inhibitors, failed to recover their size, even 48 hr post Torin-2 wash-out. This suggests that p38 inhibitors suppressed the recovery in cell size rather than slowing the kinetics associated with this process. (**C**) Cells treated with inhibitors of p38, but not inhibitors of ERK or JNK, show increased rates of proliferation as compared to control conditions, after being released from mTOR inhibition. Results shown in this figure are representative of two replicate experiments. The measurements and source code for visualization of the results is presented in *Figure 6—source data 1*.

*Figure 6 continued on next page*

*Figure 6 continued*

DOI: https://doi.org/10.7554/eLife.26947.036

The following source data and figure supplement are available for figure 6:

**Source data 1.** Cell size dynamics after released from mTOR inhibition.

DOI: https://doi.org/10.7554/eLife.26947.038

**Figure supplement 1.** Recovery in cell size is delayed even after the p38 inhibitor was wash-out.

DOI: https://doi.org/10.7554/eLife.26947.037

that disrupt the coordination of cell cycle and cell size were called off-axis compounds. p38 MAPK pathway is highly enriched for the off-axis phenotype, implicating its role in the coordination of cell size with G1 length. Follow-up experiments with specific p38 inhibitors or genetic knockdown of the pathway confirm that inhibition of p38 disturbs the coordination between size and G1 length (*Figure 2*, *Figure 3* and *Figure 4*). We further show that reduction in size results in upregulation of p38 activity (*Figure 5*), and that p38 activity is required for cells to recover their original size (*Figure 6*) by extending the duration of G1. By probing for the major regulators of G1-S transition, our results suggest that the cell cycle inhibitor p27 may play a role in how p38 affects cell cycle to maintain cell size uniformity (*Figure 2*). However, it is worth noting that, in addition to p27, previous literature has implicated several other G1 regulators as p38 effectors. It is, therefore, not unlikely that the influence of p38 on G1 length is redundantly mediated by multiple branches of signaling. In future investigations, it will be interesting to examine the specific mechanisms that explain how p38 promotes cell size uniformity and, crucially, identify how p38 'senses' cell size. Our results reveal that the p38 MAPK pathway may be involved in a 'cell-size checkpoint' that regulates G1 progression in a cell-size dependent manner. Together, this study presents the p38 MAPK pathway as a key new component of cell size control in animal cells.

Interestingly, while inhibitors of JNK or ERK/MEK also induce changes in cell size, they do not eliminate the correlation of cell size and G1 length (*Figure 2D*). Instead, these inhibitors shift this correlation to larger or smaller values of cell size. This may suggest that instead of disrupting a size checkpoint (as p38 inhibitors do), ERK/MEK or JNK inhibitors changed the target size with which that checkpoint is associated.

While our study focused on the correlation of cell size and G1 length, we also observed a weak negative correlation between G1 length and cellular growth rates, as measured by time-lapse microscopy, using nuclear size as a proxy of cell size (*Figure 3—figure supplement 1*). Such connection between cellular growth rate and G1 length has been previously reported in cultured cells with more accurate methods (*Son et al., 2012*) as well as in the context of tissues (*Datar et al., 2000*). It remains to be seen whether and how the correlation of G1 length with cell size and the correlation of G1 length with growth rate are mechanistically related.

Mechanisms by which the p38 MAPK pathway regulates cell cycle progression are well-established in literature (*Thornton and Rincon, 2009*; *Lavoie et al., 1996*; *Mikule et al., 2007*; *Yee et al., 2004*). Here, we have shown that such p38-dependent cell cycle control is regulated by cell size. Previously, it has been shown that p38 is activated by hyperosmotic conditions that shrink cell volume (*Han et al., 1994*; *Moriguchi et al., 1996*; *New and Han, 1998*). The results reported here show that the p38 pathway is activated in cells with reduced cell mass (*Figure 5*), suggesting that a common pathway responds to changes in both cell volume and cell mass. This finding is not trivial, since, unlike cell mass, cell volume is a labile phenotype that changes and recovers over rapid time scales (minutes) and is modulated by ion channels and transporters (*Lang et al., 1998*). In contrast, cell mass changes more slowly (over hours) and is modulated by protein translation and central carbon metabolism.

Given the conclusions of our study, the influence of p38 signaling on cell size in animal models is a definite point of curiosity. While literature on cell size in animal tissues is abundant (*Ginzberg et al., 2015*; *Hall et al., 2004*; *Lloyd, 2013*), focus on size uniformity or size variability is quite sparse. For example, growth in cell size has been primarily associated with activity of the mTOR pathway (*Fingar et al., 2002*). mTOR regulates cell size by promoting protein translation and metabolism (*Saxton and Sabatini, 2017*), both of which translate into cellular growth rate. In this study, we observed a decrease in size following p38 inhibition. However, this reduction in size is not

due to inhibition of mTOR, which would slow down cellular growth rates. Instead, p38 inhibition reduces cell size by shortening the duration of growth in G1 (*Figure 3*).

Defects in cell size and growth were previously implicated with genetic perturbations of p38 in mammalian tissues such as liver, bone and heart (*Tormos et al., 2013*; *Thouverey and Caverzasio, 2015*; *González-Terán et al., 2016*). Since there are four isoforms of p38 in mammals, it is interesting to ask whether these isoforms are functionally diverged with respect to their influence on cell size? Our results suggest that the coordination of G1 length with cell size depend more on p38γ and p38δ and less on p38α or p38β. Interestingly, a recent study by *González-Terán et al. (2016)* reported that mice lacking p38γ and p38δ have smaller cardiomyocytes and lower heart mass, whereas p38α knockout mice display no growth defect in heart (*Nishida et al., 2004*). The authors found knockout of p38γ and/or p38δ lead to higher level of mTOR inhibitor DEPTOR, resulting in lower mTOR activity. This may suggest that, in addition to extending cell cycle duration of small cells, p38 may also regulate growth in size by activating mTOR during development.

Last, while the present study focuses on p38 and its involvement in the regulation of cell size, we believe that the multi-parametric analysis developed in the context of the study have merit on its own. In the small molecule screen, every screened condition was assayed by single-cell measurements on thousands of unsynchronized cells, distributed over the entire cell cycle (*Figure 1B* and *Figure 1—figure supplement 2*). This single-cell data enabled detection of compounds which affect the phenotype of interest in a cell-cycle-dependent manner. Conventionally, pharmacological screens are performed to identify compounds that inhibit a particular process or activity. By contrast, the screen strategy that we developed herein sought not to inhibit a given pathway or activity but to inhibit the coordination between two biological processes (cell size and G1 length). We envision, that such an approach may have general utility. Cancer, for example, is caused not only by the mitogenic activity of an oncogene, but also by defects in cross-talk and/or feedback linking different branches of signaling. Disrupting the coordination between such disparate branches of signaling may prove important. Finally, our phenotypic screen with a biologically-annotated set of compounds (the MOA box) demonstrates the power of applying a systematic 'forward genetics' approach to chemical biology.

To conclude, in this study, we show that p38 pathway is involved in the coordination of cell size and G1 length. Aberrantly small cells display higher levels of p38 activity and grow in G1 for longer periods of time. Inhibition of p38 MAPK leads to loss of the compensatory G1 length extension in small cells, resulting in faster proliferation, smaller cell size and increased heterogeneity in size. These results suggest a model whereby the p38 MAPK pathway functions downstream of a cell-size-sensing process and feeds information about cell size to regulators of cell cycle.

## Materials and methods

### Materials

Succinimidyl ester conjugated to Alexa Fluor 647 and DAPI were purchased from Life Technologies (Burlington, ON). The following small molecule inhibitors were purchased from Selleckchem (Houston, TX): Rapamycin, Torin2, SB203580, VX-702, SB202190 (FHPI), BIRB 796 (Doramapimod), FR 180204, PD184352 (CI-1040), PD98059, JNK Inhibitor II, JNK Inhibitor IX. Lentiviral expression vectors encoding the JNK and p38 MAPK Kinase Translocation Reporters (KTR) were a kind gift from Markus Covert (Addgene plasmids No. 59151 and 59155). ON-TARGETplus SMARTpool siRNAs for the genes of interest as well non-targeting negative control siRNAs were obtained from Dharmacon (Lafayette, CO). Phospho-p38 MAPK (Thr180/Tyr182) antibody (RRID:AB_2139682) for immunofluorescence was purchased from Cell Signaling (Beverly, MA).

### Cell culture

HeLa (ATCC, RRID:CVCL_0030) and the retinal pigmented epithelial (RPE1, ATCC, RRID:CVCL_4388) cell lines stably expressing the degron of Geminin fused to Azami Green were cultured in DMEM medium (Life Technologies) supplemented with 10% Fetal Bovine Serum (FBS, Wisent, Montreal, QC) at 37°C in a humidified atmosphere with 5% $CO_2$. RPE1 cells stably co-expressing H2B conjugated to mTurquoise and Geminin conjugated mVenus were cultured in DMEM/F12 medium (Life Technologies) supplemented with 10% FBS. RPE1 cells stably expressing the JNK or p38 MAPK

KTRs were generated as described below. Briefly, Human embryonic kidney HEK293T (ATCC, RRID: CVCL_0063) cells were maintained in DMEM (Wisent) supplemented with 10% FBS at 37°C and 5% $CO_2$. The lentiviral transfer plasmids encoding the KTRs were co-transfected with plasmids encoding the packaging genes (viral Gag-pol) and the envelope gene, VSV-G, into the packaging HEK293T cells using jetPRIME transfection reagent (Polyplus Transfection New York, NY). The medium was changed 24 hr post-transfection and the viral supernatants were harvested 48 hr later, passed through a 0.45-μm syringe filter and frozen at −80°C. The retrovirus was thawed at room temperature and then used to transduce RPE1 cells with the respective lentiviral transduction particles. Resistant clones were selected in 4 μg/mL puromycin (InvivoGen, San Diego, CA) for 3 days, isolated using cloning cylinders, and subsequently expanded and maintained in puromycin-containing DMEM medium. In all experiments, cell density was monitored to avoid over-confluence and contact inhibition. All cell lines used in this study have been tested for and confirmed to be negative for mycoplasma contamination.

## Compound screen

Two internal Novartis sets of tool compounds were screened, the publicly known subset of compounds in the Mechanism-of-Action (MOA) Box (1609 compounds) and a Kinome Box (1637 compounds). The MOA Box is a dynamically managed annotated list of compounds that cover target, pathway, and bioactivity space as comprehensively as possible to facilitate biological discovery by screening and profiling experiments. Bioactivity annotations were derived from integrated in-house assays and external assay data sources containing a mixture of qualitative target assignments (Thomson Reuters Integrity, DrugBank, Novartis-nominated MOA Box members) or quantitative dose response-type experimental assay data (chEMBL, GVK GOSTAR, Novartis assay data). Compounds were prioritized per target based on availability, amount of target evidence and, if available, clinical phase, while limiting the number of compounds selected from any one assay, publication, or patent. At the time of screening, chemical modulators of 903 unique human primary targets were represented based on hand-annotation, typically by 1–5 compounds/target where a compound-target association was made. In total, 42% of MOA Box compounds have only one assigned primary target, and a mean of 2 primary targets. However, additional target coverage for MOA Box compounds could be inferred from, the integrated sources listed above, comprising 1964 total unique targets, such that 11% of members have only one inferable target and a median of 6 inferable targets. Target enrichment analysis, described below, was carried out using all possible targets (assigned and inferred from data). The target class coverage by percent of the MOA Box is as follows (*Figure 1—figure supplement 1*): Non-kinase Enzymes (28%), Kinases (16%), GPCR (15%), Proteases (12%), Ion Channel (11%), Unclassified (4%), Transporters (3%), Other Receptors (3%), Nuclear Hormone Receptors (3%), Cytokines (3%), and <1% each of Transcriptional Regulators, Signal Transducers and Structural Molecules. To assemble the Kinome Box, all physically-available inhibitors with publically-known structures having IC50 values < 1 μM for a human kinase were filtered for only those inhibiting <25 total kinases in integrated internal and external data. A total of 473 human kinases were covered by 1637 compounds (mean and median number of targets per compound equal to 5.4 and 2, respectively).

The screen was performed in 384-well μclear microplates (Greiner Bio-one, Monroe, NC). On day 1, 2000 cells per well were seeded in 30 μl medium. On day 2, compound was added to a final concentration of 1 μM, 3 μM or 10 μM (or 2 μM, 6 μM or 20 μM for Kinome Box) from a 2 mM or 10 mM stock solution using a Biomek FX with 384-well head (Beckman Coulter, Indianapolis, IN). The DMSO concentration was kept below or at 0.5% v/v. To reduce stochastic noise and promote overall screen accuracy, each compound was screened with three concentrations and duplicates.

## Compound treatment

The cells were seeded into 96-well μclear microplates (Greiner Bio-one, Monroe, NC) 16 hr prior to drug treatment. To perturb the p38, JNK and Erk pathways, the following specific inhibitors were used at the concentrations indicated in the figure legends. All compounds were dissolved in DMSO and diluted in DMEM when used. The concentrations have been selected to avoid severely interfering with cell proliferation or cause noticeable cell death. Twenty-four hours post compound treatment (control wells were treated with DMSO), the cells were treated and incubated with Rapamycin

(30, 3, 0.3, 0.1 and 0.03 nM) for an additional 24 hr before fixation and staining, or for whole cell lysis. DMSO only (0.01–0.5% v/v in DMEM) was used as a negative control for all the compound treatment experiments.

To search for an appropriate working concentration of the MAPK inhibitors, we tested a series of different concentrations starting with the compounds IC50, as published by the vendors. However, it is worth noting that the IC50 of a given compound may vary from one cell line to another. Also, many reported IC50 are based on cell-free assays which highly depend on assay conditions and are usually much lower than the working concentrations that are relevant for cell culture. As an example, the reported IC50 (cell-free assay) of SB203580 on p38 ranges from 0.19 nM (*Gaestel et al., 2007*) to 7 µM (*Wei et al., 2007*). Because of these inherent inaccuracies in published IC50s, a range of concentrations for each of the chemical inhibitors were tested. To start, we performed four serial dilutions of each compound, where the lowest tested concentration was the compounds published IC50, as advertised by the vendor website. Compound concentrations that resulted in a significant decrease in cell number (e.g. half the cell number of the control wells) were deemed toxic and excluded from following analyses. For example, the compound PD184382 resulted in decreased cell count when used at concentrations higher than 300 nM. Conversely, if a given concentration (of a compound) resulted in a phenotype that is indistinguishable from wild-type, the entire concentration series for that specific compound was recalibrated to start at a higher concentration value. Examples of the latter include the compounds, TAK-715 and JNK inhibitor II. Using this procedure of calibration, we selected a specific working concentration for each of the compounds used in the study.

To examine the process of recovery in cell size, cells were treated with Torin2 (50 and 25 nM) for 24 hr, washed with PBS and re-incubated with fresh media for the times indicated. Cells were resuspended using trypsin and cell size and cell density was measured using the Multisizer 4 Coulter Counter (Beckman-Coulter, Mississauga, ON) or collected for whole cell lysis at time points indicated in the figures.

In the experiment using p38 or JNK KTRs, cells were seeded in 96 well plates and treated with rapamycin (30, 3, 0.3, 0.1, 0.03 nM and DMSO controls) for 24 hr. The cells were either fixed and stained or washed with PBS and replaced with fresh DMEM media for 6 hr before fixation and staining.

## siRNA transfection

RPE1-Geminin cells were seeded in 6-well plates at a density of $2 \times 10^5$ cells/well. Twenty four hours post-seeding, the cells were transfected with SMARTpool siRNA (25 nM) using DharmaFECT one transfection reagent according to the manufacturer's instructions. Twenty hours post transfection, the cells were re-suspended using 0.05% trypsin-EDTA (Wisent) and re-seeded into 96-well µclear microplates (Greiner Bio-one, Monroe, NC) at a density of 5,000 cells/well. Six hours after re-seeding, cells were treated with rapamycin (30, 3, 0.3, 0.1 and 0.03 nM) or DMSO control for 24 hr before fixation and staining. The experimental procedures were optimized so there is no observable cell cycle arrest or cell death. Specifically, the siRNA-transfected cells were washed-out of transfection reagent and cultured in regular media when used for the assay with rapamycin concentration series. Cells were observed to continuously increase in number during the assay and proliferate with a similar rate as non-transfected cells.

## Immunostaining

The procedures followed the Cell Signaling protocol. Briefly, cells were first fixed with 4% PFA for 15 min, blocked in blocking buffer for 1 hr. Afterwards, samples were incubated in primary antibody at 4°C overnight, and subsequently secondary antibody for 1 hr before imaging.

## Fixation, staining and imaging

Cells were fixed in 4% paraformaldehyde (Electron Microscopy Sciences, Hatfield, PA) for 10 min, followed by permeabilization in cold methanol at −20°C for 5 min. Cells were stained for cell size with 0.4 µg/mL Alexa Fluor 647 conjugated succinimidyl ester (SE) for 2 hr to nonspecifically label total protein content. The cells were then labeled for DNA with 1 µg/mL DAPI for 10 min. The cells were either imaged using an IN Cell Analyzer 2000 HCA system (GE Healthcare Life Sciences, Pittsburgh, PA) microscope at 10X magnification (compound screen) or using the Operetta High-Content

Imaging System (Perkin Elmer, Woodbridge, ON) at 20X magnification (compound treatment and time-lapse experiments).

## Whole cell lysis and western blotting

To prepare whole cell lysates, cells were rinsed with ice-cold PBS and solubilized with RIPA Lysis Buffer (Boston Bio-Products, Boston MA) [50 mM Tris-HCl, 150 mM NaCl, 5 mM EDTA, 1 mM EGTA, 1% NP-40, 0.1% SDS and 0.5% sodium deoxycholate, pH 7.4] supplemented with protease and phosphatase inhibitor Cocktail (Thermo Scientific, Burlington, ON). Protein concentration was determined using the BCA protein assay (Thermo Scientific, Burlington, ON) and suspended with 4X Bolt LDS Sample Buffer and 10X Bolt Reducing Agent and heated for 10 min at 70°C. Samples of equal protein were resolved by SDS-polyacrylamide gel electrophoresis and subjected to immunoblotting for proteins as indicated. The antibodies used for immunoblotting were all purchased from Cell Signaling Technology (Beverly, MA). All western-blot results in the figures have been reproduced in replicate experiments with cell lysates samples prepared in independent experiments.

## Image processing and cell segmentation

Automated image-processing pipelines have been developed using Matlab (RRID:SCR_001622). The general scheme includes three steps: correction for background fluorescence and/or uneven illumination, cell segmentation, and feature quantification. Each step has been optimized according to microscopes, experimental design (e.g. fixed or live cell) and the features needed. The same image processing pipeline and parameters were applied identically to all images from the same experiment.

To correct for background fluorescence, a background image was constructed and applied per channel. There are two scenarios to construct a background image: either by averaging images containing few or no cells (in the compound screen); or by averaging the background region across all images collected (experiments with KTR cells, and in the time-lapse imaging). Uneven illumination was corrected for images from the compound screen in which the problem is obvious. A flat-field image has been constructed and applied for all channels. The flat-field image was constructed based on the fact that 2N peak in DNA distribution is invariant to cell positions in the image.

In the step of cell segmentation, nucleuses were first spotted through a nuclear channel (DNA or H2B), and segmented by seed-based watershed. When cell channel exists, the segmented nucleuses were further used as seeds to segment cells (SE channel) by watershed. The nuclear and cell border were detected with thresholds in corresponding channels, which were either automatically detected or manually selected based on need.

Following segmentation, each individual cell was processed to collect for its quantitative features, including total/average fluorescence per channel, nuclear size, *etc*.

To calculate the activity of the KTR, the fluorescent intensity of the KTR channel was collected and integrated within either the cytoplasmic region (identified by SE channel) or the nuclear region (identified by DAPI channel) to calculate the ratio of cytoplasmic to nuclear intensity. Note the integrated intensity correlates with the amount of fluorescent protein while intensity/brightness is usually used as an estimate for protein concentration. Further, measurements with integrated intensity is more robust to morphological changes, while brightness measured with epiflourescence could change dramatically depending on the thickness of the cell. As the experiment used in this study undergo extended treatment (over 20 hr) and the cells could change their morphology under treatment, it is more reasonable to use integrated intensity to estimate shuttling of proteins.

## Cell cycle stages

Cells were first partitioned, according to the nuclear DNA level, into G1 (2N), S (2 N-4N) and G2 (4N) phase. Progression in G1 phase was further divided, based on fluorescence of nuclear Geminin (and nuclear Cdt1 if available) into early G1 (low Geminin), late G1 (medium Geminin, high Cdt1), and G1/S transition (higher Geminin, medium to high Cdt1). The thresholds were automatically detected based on distribution of DNA, log(Geminin) and log(Cdt1).

## Estimation of cell cycle durations and growth rate from bulk measurements

Cells were treated with inhibitors and fixed every 20 hr over a period of 3 days. The rate of cell proliferation and protein accumulation were calculated by fitting a log model (assuming that cells proliferate and increase protein content exponentially). To calculate the duration for each cell cycle stage, the cells were first portioned into G1, S and G2 to calculate the proportion of cells in each cell cycle stage. The duration of a cell cycle stage is then calculated with the following formula: $t = T \cdot \frac{PDF}{2 - CDF}$, where $t$ is the duration of the specific cell cycle stage, PDF is the proportion of the cells in that stage, and CDF is the proportion of cells in or earlier than that stage. This formula is a simplification of the ERA method (*Kafri et al., 2013*).

## Analysis of the compound screen

The analysis was performed per plate separately, due to observed variability among plates. After image-processing of the compound screen datasets, every well captures approximately 4000 cells. For each well, both cell size at early G1 stage (*S*) and proportions of cells in early G1 stage (*P*) were collected. The minimum volume ellipsoid (MVE) estimator (*Rousseeuw, 1985*; *Rousseeuw and van Zomeren, 1990*) was applied to pick a smallest volume ellipsoid that consists of around 50% of the total sample. The MVE estimator was a reliable tool to identify robust trend and detect outliers in multivariate data. The subsample of minimum volume ellipsoid was used to calculate the robust covariance matrix (rCov), from which correlation coefficient between *S* and *P* was calculated per plate. Further, we calculated the Mahalanobis distance from each data point (a well) to the center of the ellipsoid, and performed thresholding (5% significance) to the Mahalanobis distance to detect the outlier wells. Intuitively, the outlier wells display drastic distinctive phenotype in early G1 cell size and/or its G1-length compared with the control wells (and most drug treatments). Based on rCov, the principle vector of [*S*, *P*] was calculated and the angle of each outlier well with reference to the major principle vector was computed. The outlier angles were used to classify on/off-axis outliers: outliers with an angle smaller than 45 degree were classified as on-axis outliers, and otherwise off-axis outliers. The on/off-axis outliers were further filtered separately to lower false-positive rate. In the compound screen, each compound has been tested for at least six times (three concentrations with duplicates); and compounds that have only been identified as outlier once among all treatments were excluded from further analysis. Next with the robust outliers, we performed the target enrichment analysis using hypergeometric test (Fisher's exact test) for each known target of the outliers to identify the target proteins that are significantly highly represented in the outlier compounds.

To identify compound treatments that increase cell size variability, median absolute deviation (MAD), a robust measure of variability, of cell size was calculated per well. We observed that cell size MAD is linearly correlated with median cell size ($r = 0.964$). Accordingly, we normalize the MAD by median cell size to obtain a robust measure of cell-to-cell variability within cell populations. For each well, its normalized MAD is compared to that of the control wells in the same plate to calculate a MAD score. To select for compound treatments with perturbed size variability, a threshold in MAD score was calculated based on the distribution of MAD scores of all control wells (1% significance). The outliers with higher MAD scores compared with the threshold were filtered for repeatability and enriched for target proteins the same way as described above.

## Time-lapse microscopy

RPE1 cells with stable expression of H2B-mTurquoise and Geminin-mVenus were seeded in 96-well μclear microplates (Greiner Bio-one, Monroe, NC) and grown in the incubator for at least 6 hr prior to imaging. The cells were imaged with the Operetta High-Content Imaging System. During the imaging, the plate was incubated in the live cell chamber (37°C, 5% $CO_2$) and grown in the Fluoro-Brite DMEM supplemented with FBS, L-glutamine and Sodium Pyruvate. As the cells were previously cultured in regular DMEM and displayed suboptimal cell proliferation after switching to FluoroBrite DMEM, the cells were grown in FluoroBrite medium for a period of 2 weeks to adapt to the new medium before the time-lapse experiments. Widefield fluorescent images of H2B-mTurquoise and Geminin-mVenus were collected every 15 min at 20X magnification for 50 hr. Under this experimental setting, the microscope could support imaging of up to four wells every time.

## Automated lineage tracking and analysis

The live-cell images were first processed with the cell-segmentation pipeline to obtain single-cell features including cell position, fluorescent intensity and nuclear size. Subsequently, cells from each time point $T$ were compared with the ones from $T + 1$ to track for cell motion and division by searching for the globally optimal matches between neighboring time points (*Kanade et al., 2011*). Parameters used for tracking were automatically calculated based on distribution of cell features and confirmed in subsamples by eye. As output from tracking, each track starts with either appearance (first time point, or cell move into image field) or cell birth (with known mother and sister cell), and end with disappearance (last time point, or cell move out of image field) or cell division (with known daughter cells). After tracking, the individual cell tracks were further filtered to select for accurate tracks with the full cell cycle captured. Specifically, the algorithm selects for cell tracks that start with typical features of cell birth, end with typical features of mitosis and have relatively smooth fluorescence dynamics. Further, for each full cell cycle track, swift rise in Geminin level was detected to quantify early G1 duration. Initial nuclear size is estimated by averaging the first nine data points (~2.5 hr) after cell birth to decrease noise. Similarly, nuclear size at G1/S transition is estimated by averaging the ±4 data points (~2.5 hr) at time point of G1/S transition. Nuclear size measurements collected during mitosis were excluded from analysis, as H2B-mTurquoise does not accurately depict nuclear size during mitosis when nuclear envelope breaks down and chromosome condenses. To calculate growth rate in nuclear size, dynamics of individual nuclear size over entire cell cycle was first fitted with a smoothing spline model. The dynamics of the growth rate is then calculated with the first derivative of the fitting result.

## Acknowledgements

We thank the Canadian Institutes of Health Research for its grant to RK (FRN-343437), and the National Institute of General Medical Sciences for its grant to MK (GM26875). We also thank Patricia and Alexander Younger and the Younger foundation for their generous donation to support this research. SL was supported by a graduate studentship award from the Research Training Center at the Hospital for Sick Children.

## Additional information

### Competing interests

Marc Hild, Yuan Wang, Jeremy L Jenkins: Employee at Novartis Institutes for Biomedical Research who had no role in study design, data collection and interpretation, or the decision to submit the work for publication. The other authors declare that no competing interests exist.

### Funding

| Funder | Grant reference number | Author |
| --- | --- | --- |
| Canadian Institutes of Health Research | FRN-343437 | Ran Kafri |
| National Institute of General Medical Sciences | GM26875 | Marc W Kirschner |

The funders had no role in study design, data collection and interpretation, or the decision to submit the work for publication.

### Author contributions

Shixuan Liu, Conceptualization, Data curation, Software, Formal analysis, Validation, Investigation, Visualization, Methodology, Writing—original draft, Project administration, Writing—review and editing; Miriam Bracha Ginzberg, Conceptualization, Software, Investigation, Methodology, Writing—review and editing; Nish Patel, Conceptualization, Supervision, Validation, Investigation, Visualization, Methodology, Writing—review and editing; Marc Hild, Jeremy L Jenkins, Conceptualization, Resources, Data curation, Supervision, Investigation, Methodology, Project administration; Bosco

Leung, Ceryl Tan, Formal analysis, Investigation; Zhengda Li, Software, Formal analysis, Investigation, Visualization, Methodology; Yen-Chi Chen, Formal analysis, Validation, Investigation, Methodology; Nancy Chang, Shulamit Diena, Validation, Investigation; Yuan Wang, Data curation, Investigation, Methodology; William Trimble, Supervision, Investigation; Larry Wasserman, Supervision, Investigation, Methodology; Marc W Kirschner, Conceptualization, Resources, Supervision, Funding acquisition, Project administration, Writing—review and editing; Ran Kafri, Conceptualization, Resources, Data curation, Software, Formal analysis, Supervision, Funding acquisition, Investigation, Methodology, Project administration, Writing—review and editing

## Author ORCIDs

Yen-Chi Chen ⓘD http://orcid.org/0000-0002-4485-306X
Marc W Kirschner ⓘD https://orcid.org/0000-0001-6540-6130
Ran Kafri ⓘD http://orcid.org/0000-0002-9656-0189

## Decision letter and Author response

Decision letter https://doi.org/10.7554/eLife.26947.041
Author response https://doi.org/10.7554/eLife.26947.042

## Additional files

**Supplementary files**

• Transparent reporting form
DOI: https://doi.org/10.7554/eLife.26947.039

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
