## [Decision Letter]

Thank you for submitting your article "Size uniformity of animal cells is actively maintained by a p38 MAPK-dependent regulation of G1-length" for consideration by *eLife*. Your article has been reviewed by three peer reviewers, one of whom is a member of our Board of Reviewing Editors, and the evaluation has been overseen by Marianne Bronner as the Senior Editor.

While the three reviewers agreed that the paper addresses an important topic and reports some very relevant and potentially important results in identifying P38 in size sensing, reviewers 1 and 3 raised a number of very substantial criticisms. Some of their queries relate to unclear presentation, whereas others relate to data that apparently don't support the paper's conclusions. These of course must be dealt with.

The Reviewing Editor has drafted this consolidated review to help define the key issues we believe must be addressed before the Board and reviewers can make a binding decision.

Summary:

In this manuscript, the authors probe the molecular regulation of cell size regulation during G1 phase. Starting from a small molecule screen, they identify the p38 MAPK pathway as important to link cell size with cell cycle progression. By modulating cell size through TOR inhibition, they show that p38 and upstream regulatory kinases are required for G1 phase extension in small cells. They propose that p38 becomes activated in response to changes in size and controls G1 duration to ensure cell size uniformity. Cell size homeostasis is very poorly understood in nearly all cell types. Thus, the finding that p38 MAPK plays an important role is a first molecular entry into understanding cell size regulation at molecular level in animal cells. This is likely to be an important finding.

Essential revisions:

1) The text, Materials and methods, and figure legends lack important details necessary to interpret the data shown. What are all the points in the graphs (Figure 2, Figure 3, Figure 4). Are these replicates or a time series of concentration series? How many cells make up each data-point?

2) Figure 2 is very difficult to read. The size of the dots in panels A and B is not explained. The difference between data shown in panels A and B (non-SB203580-treated) is not clear.

In panel D, the choice of concentration of each of the inhibitors is not explained. It would make sense to use concentrations that correspond to the IC50 for each inhibitor, or fraction thereof, such that different inhibitors can be compared (i.e. at `low dose` the chosen concentrations vary from known 3 to 100 fold the known IC50 for these compounds). This is especially important for compounds showing no phenotype – one needs to be sure that the pathway is indeed inhibited at the concentration used. It would be also necessary to state that off-target effects are also possible, and thus p38 is one candidate regulator but others are not excluded. Especially for the high dose treatment used for SB203580, PKB could potentially be inhibited. In this panel, it would also help to color each dot to indicate the different rapamycin concentrations used, as in panel A.

3) The P38 inhibitors make the cells very, very small, but this isn't discussed. Is their growth rate changed? Division rate? It's not until Figure 4—figure supplement 1 that some data on this important fact is shown. A complete cell cycle analysis of these cells seems essential. Are they delayed in G2? What is their growth rate and division rate?

4) The western blots shown in panel 2C are not convincing and do not support several of claims made in the text. p-p38 doesn't increase with TOR inhibition, and worse, it doesn't decrease with the P38 inhibitor. This is a critical flaw. We do not see upregulation of p27. We do not see evidence that p38 inhibition disrupts the effect of rapamycin on the expression levels of cell cycle regulators. We also do not see that p38 is activated when cell size is decreased by inhibition of mTOR. In summary, these data should be obtained in several repeats (at least 3), quantified, and statistics shown. The data as shown raise grave doubts as to the paper's conclusions.

5) In Figure 4 and supplements, I do not understand the rationale for performing measurements over 48h after Torin removal, when the authors showed in panel A-B that the meaningful time to look at size-altered cells is between 1 and 20 hours post-Torin wash-out. In 4D, the only timepoint that falls within this interval is the 6h timepoint, and it does not look like this one is different from the control upon p38 inhibition. Thus I am left unconvinced by this data.

6) The logical flow of the paper is clear and compelling but with the one caveat other explanations might be found for the delayed kinetic response of p38 inactivation after mTOR inhibitor washout. For instance, it could simply be that the kinetics of p38 inactivation are slower than for mTOR recovery, i.e., that the small size is not directly the cause of p38 activation. In other contexts, how fast is p38 inactivated after a stimulus is removed? I don't see an obvious experiment to address this issue.

7) The raw screen data does not appear to be reported in a supplementary table so it is not possible for the analysis to be confirmed, or for further information to be extracted from the data after publication. The raw data for each compound in the screen should be explicitly provided in a supplementary table so it is easy for readers to access.

8) The authors state that loss of p38 increases cell size variability, but little if any data is provided that directly support this important, central point.

[Editors' note: further revisions were requested prior to acceptance, as described below.]

Thank you for resubmitting your work entitled "Size uniformity of animal cells is actively maintained by a p38 MAPK-dependent regulation of G1-length" for further consideration at *eLife*. Your revised article has been favorably evaluated by Marianne Bronner as Senior editor and two reviewers, one of whom is a member of our Board of Reviewing Editors.

The manuscript has been improved but there are some remaining issues that need to be addressed before acceptance. As you can see from the reviews, appended below, the reviewers appreciated the careful revisions you submitted, and agreed that the paper reports an important new discovery about cell size control and cell cycle progression that warrants publication. They felt the presentation was much improved, and found the central data on P38 convincing. The reviewers do include several comments about how the paper could be improved further, and I encourage you to address these in the final revision. However further data from new experiments is not required at this stage. Thanks for your patience with the review process – I must apologize for it taking so long.

Reviewer #1:

As before, this paper reports a detailed analysis of the relationship between cell size and cell cycle progression that includes a large screen for compounds that disrupt this relationship. The methods for studying this relationship are innovative, high throughput, and accurate, and the screen convincingly pulled out inhibitors of P38-MAPK as agents that specifically disrupt the normal proportional relationship between cell size and cell cycle stage. The revision is much improved. It is now more concise and easy to read, and contains many important details of methods that were previously missing. Some distracting, irrelevant data has been removed and a few additional tests and analyses have been added that enhance the presentation. The central points, namely that cell size and cell cycle stage are normally coupled in human cells, and that loss of P38 can sever this coupling, are very well supported. This function of P38 appears to be somewhat unique, as well as unexpected, and warrants a good report in the public literature. However the paper still could be more compelling if more were reported about how p38 senses normal size/cell proportionality, and how it affects the cell cycle to maintain size uniformity. With regards to this latter point, the authors suggest it may work via the cyclin kinase inhibitor p27, which show's levels changes consistent with a regulatory function. However the p27 expression data here is cursory, and it's function in size maintenance is never tested, which is unfortunate. I would like to encourage the authors to perform a few functional tests of P27, since demonstrating that it's response is meaningful could enhance the paper a lot.

Reviewer #2:

In my opinion, the authors have addressed all main issues raised by the reviewers. The manuscript is much more clearly written, and in particular the logical flow and main observations are now explained in sufficient detail to easily understand the thrust of the manuscript. To reiterate my main positive sentiments, the recognition that off-axis compounds are informative for size coupling represents a significant insight into size control, and stands in marked contrast to most previous genetic analysis of cell size that revealed on-axis pathways (PI3K, Akt, mTOR). This argument is convincingly exemplified for p38, as many compounds implicated in p38 regulation cause off-axis effects and increases in size variability. Appropriate consideration is now given to issues of chemical inhibitor specificity, and additional genetic controls clearly implicate the p38 kinases in G1 phase-specific size regulation. The mTOR inhibitor cell size recovery experiments are also much better explained and supported by additional controls. While the manuscript does not identify the precise mechanisms whereby p38 regulates size, a tantalizing correlation with p27 levels is provided, which should form the basis for interesting follow-on studies. Overall, this is a compelling manuscript that illuminates p38 as a key new component of size control in human cells.

---

## [Author Response]

Essential revisions:1) The text, Materials and methods, and figure legends lack important details necessary to interpret the data shown. What are all the points in the graphs (Figure 2, Figure 3, Figure 4). Are these replicates or a time series of concentration series? How many cells make up each data-point?

In the revised manuscript, we added detailed descriptions to the legends of each of the figures mentioned in the referee’s comments above. In addition, we added explanatory details to the main article text to make the experimental procedures and figures as clear as possible.

As to the referee’s specific question. We quote from the revised figure legends of Figure 2:

“Samples of unsynchronized cells were treated with increasing concentrations of rapamycin (a rapamycin concentration series) for a period of 24 hours, and then stained and imaged to quantify cell size and cell cycle stage on a single cell basis. […] Each data point was measured from an unsynchronized population with a minimum of 7,000 cells.”

The number of cells that make up each data-point differs by experiment and treatments. The specific cell number for each experiments is indicated in the corresponding figure legends (minimal 3000 cells).

We also improved the representation of these figures by using circle size to discriminate different rapamycin concentrations. Consistent in all figures, larger circle size indicates a higher rapamycin concentration.

2) Figure 2 is very difficult to read. The size of the dots in panels A and B is not explained. The difference between data shown in panels A and B (non-SB203580-treated) is not clear.

We revised the visualization of Figure 2 as indicated in response to major point #1. The size of the dots is used to distinguish different concentrations of rapamycin treatment.

In the revised Figure 2, panel A is dedicated solely to demonstrate and describe the experimental assay that we developed to quantify the correlation of cell size and G1-length. Panel B of Figure 2 implements this assay to test how p38 inhibition (red points) influences the correlation of cell size and G1 length, as compared to control condition where p38 is not inhibited (blue points).

We revised the legend for Figure 2 to improve clarity, as quoted below:

“The experiment described in panel A is repeated with (red) or without (blue) a chemical inhibitor of p38 (SB203580, 5 μM). The negative correlation between the size of early G1 cells and the proportion of cells in G1 is apparent in populations not treated with SB203580 (blue) but not in the populations that are treated with SB203580. The blue and red trend lines represent linear regressions.”

In panel D, the choice of concentration of each of the inhibitors is not explained. It would make sense to use concentrations that correspond to the IC50 for each inhibitor, or fraction thereof, such that different inhibitors can be compared (i.e. at `low dose` the chosen concentrations vary from known 3 to 100 fold the known IC50 for these compounds). This is especially important for compounds showing no phenotype – one needs to be sure that the pathway is indeed inhibited at the concentration used. It would be also necessary to state that off-target effects are also possible, and thus p38 is one candidate regulator but others are not excluded. Especially for the high dose treatment used for SB203580, PKB could potentially be inhibited. In this panel, it would also help to color each dot to indicate the different rapamycin concentrations used, as in panel A.

We accept the referee’s criticism. In the revised manuscript, we now include detailed description in the Materials and methods section (Compound treatment) on how we chose the specific concentrations of the chemical inhibitors used in the study. Below is a quote from the revised Materials and methods section that addresses this concern:

“To search for an appropriate working concentration of the MAPK inhibitors, we tested a series of different concentrations starting with the compounds IC50, as published by the vendors. […] Using this procedure of calibration, we selected a specific working concentration for each of the compounds used in the study.”

To demonstrate the quality of our selected concentration series, we include a new revised supplementary figure showing the dose-dependent influence of p38 inhibitors on cell size and on the proportion of cells in G1, for all 4 tested concentrations (Figure 2—figure supplement 3).

The reviewer has suggested two possible concerns regarding the choice of compound concentrations.

“… compounds showing no phenotype – one needs to be sure that the pathway is indeed inhibited at the concentration used.” We agree with the referee’s suggestion. We used western blots to confirm that the inhibitors indeed inhibit their target pathways at the concentrations used (Figure 2—figure supplement 1). In addition, while our experiments show that chemical inhibitors of Erk and JNK did not disrupt the correlation of cell size and G1-length, these inhibitors did exert significant influence on cell size (Figure 2, compare the red points to the blue points on cell size axis). This confirms that while these compounds had little or no influence on one phenotype (the coordination of size and G1), they did display an influence on other phenotypes, e.g. cell size. To preemptively avoid confusion, we stress in the article main text that while these inhibitors increase or decrease cell size, the coordination with of size and G1 length is maintained under their influence, as quoted below:

“By contrast, inhibitors of JNK or ERK/MEK, do not significantly weaken the correlation of size and G1 length. Instead, these inhibitors shift the correlation to larger or smaller values of cell size (Figure 2).”

“It would be also necessary to state that off-target effects are also possible, and thus p38 is one candidate regulator but others are not excluded. Especially for the high dose treatment used for SB203580, PKB could potentially be inhibited.” We agree. Off-target effects are a real concern when using chemical inhibitors. To confirm that the observed phenotypes are, indeed, unique to p38 and not the result of off-target effects, we relied on two parallel experimental strategies. One, using a number of structurally different chemical inhibitors rather than relying on one single inhibitor. Two, using genetic perturbations to validate the predictions of the chemical perturbations. Specifically, we quote from the revised version of the manuscript:

“Our results show that, consistently, inhibitors of p38 disrupt the negative correlation of size and the proportion of cells in G1 (Figure 2, Figure 2—figure supplement 1, 2 and 3). This result was observed with multiple chemical inhibitors of p38 that are distinct in chemical structure (Gaestel et al., 2007), making it less likely that the observed phenotype is owed to off-target effects.”

Also, to further alleviate the concern of artifacts caused by chemical inhibitors, we additionally validated our conclusions by genetically knocking down p38 pathway components with siRNAs (Figure 4).

3) The P38 inhibitors make the cells very, very small, but this isn't discussed. Is their growth rate changed? Division rate? It's not until Figure 4—figure supplement 1 that some data on this important fact is shown. A complete cell cycle analysis of these cells seems essential. Are they delayed in G2? What is their growth rate and division rate?

We thank the reviewers for this advice. In the revised manuscript, we include detailed discussion on the influence of p38 on cell size. Also, we provide new data from comprehensive cell cycle analyses, examining how inhibition of p38 influences each of the different cell cycle stages. To assist the referees, we provide below a quote from the revised manuscript that describes these new results.

“In addition to weakening the correlation of G1 length and cell size, p38 inhibitors also cause a reduction in average cell size (Figure 2). […]For each collected sample, we measured the total bulk protein mass and total cell count, which were then used to estimate the average rate of cell growth and the average rate of cell division (see Materials and methods – Estimation of cell cycle durations and growth rate from bulk measurements).

As shown in Figure 3,7 inhibition results in shorter cell cycles (Figure 3, Figure 3—figure supplement 3), with a consequential increase in rates of cell division (Figure 3). In addition, the results show that the shortened cell cycle length caused by p38 inhibitors is owed to a decrease in G1 length (Figure 3 and Figure 3—figure supplement 3) with no significant changes in the lengths of S phase or G2 (Figure 3, Figure 3—figure supplement 3 D and E). At G1/S transition, cells subject to p38 inhibition have smaller nucleus size as compared to control (Figure 3). This decrease in cell size is caused by a shorter growth duration (in G1) rather than a slower growth rate. This conclusion is independently demonstrated by each of the two experiments described above, both showing similar growth rates for p38-inhibited cells and control cells (Figure 3—figure supplement 3, Figure 3—figure supplement 1). Altogether, this suggests that the p38 pathway mediates a cell size checkpoint at the G1 transition, selectively allowing large but not small cells to progress into S phase. Potentially, this model also explains why p38 inhibitors reduce cell size. By disrupting the cell size checkpoint, p38 inhibitors cause cells to progress through the G1/S transition with prematurely small size, causing an overall reduction in cell size.”

4) The western blots shown in panel 2C are not convincing and do not support several of claims made in the text. p-p38 doesn't increase with TOR inhibition, and worse, it doesn't decrease with the P38 inhibitor. This is a critical flaw. We do not see upregulation of p27. We do not see evidence that p38 inhibition disrupts the effect of rapamycin on the expression levels of cell cycle regulators. We also do not see that p38 is activated when cell size is decreased by inhibition of mTOR. In summary, these data should be obtained in several repeats (at least 3), quantified, and statistics shown. The data as shown raise grave doubts as to the paper's conclusions.

We thank the referees for identifying this point of confusion. The confusion that the referee points out is the result of our failure to appropriately explain our experiment and its expected outcomes. In the revised manuscript, we now added a new section that clarifies and explains this better. In the revised manuscript, this new section is under the title: p38 mediates the size-dependent G1 length extension by influencing G1/S regulators. Briefly, the western blots that the referee’s comment address were designed to test which cell cycle regulators mediate the influence of p38 on cell cycle progression.

To address the specific question of the reviewer, we now include in the legend of Figure 2 of the revised manuscript, an explanation as to why p-p38 does not decrease with the p38 inhibitor. For the convenience of the editors and referees, we add below a quote from this revised section:

“The increased levels of phopho-p38 in the population that is treated with SB203580 (a p38 inhibitor) should not be interpreted as a lack of efficacy of SB203580. Rather, these higher levels of phopho-p38 are explained by a negative feedback in the p38 pathway (Arthur and Ley, 2013), and the fact that while p38 inhibitors prevent p-p38 from phosphorylating its downstream substrates, these inhibitors do not block phosphorylation of p38 itself by upstream regulators (Kumar et al., 1999).”

As suggested, we repeated this western-blot experiments 3 times and quantified how inhibition of mTOR pathway and/or p38 pathway affects levels of p-p38, p-CREB (downstream of p38) and p27. The results of this quantification have been added as a new supplementary figure (Figure 2—figure supplement 4). In addition to the experiments with rapamycin, we now included new experiments with Torin-2, a more potent mTOR inhibitor that decrease cell size to greater extents. As a result, the new manuscript includes a new Figure 2, which now displays results of both rapamycin and Torin-2. To assist the editors and referees, we present below a quote from the revised manuscript that describes these new results:

“To test the mechanisms of the p38-mediated cell size checkpoint, we treated cells with inhibitors of mTOR, rapamycin and Torin-2, to induce a size-dependent lengthening of G1. […]We suggest that this may be the mechanism explaining how inhibition of p38 disrupts the compensatory G1 lengthening in small cells.” (From Results section – p38 mediates the size-dependent G1 length extension by influencing G1/S regulators)

Also described in the Figure 2—figure supplement 4 legend: “Treatment of rapamycin or Torin-2 increases both p-p38 and p-CREB, confirming that activity in the p38 pathway is upregulated under mTORC1 inhibition. […] This may be the mechanism by which p38 inhibition disturbs the cells’ ability to compensate their small size with longer G1.”

5) In Figure 4 and supplements, I do not understand the rationale for performing measurements over 48h after Torin removal, when the authors showed in panel A-B that the meaningful time to look at size-altered cells is between 1 and 20 hours post-Torin wash-out. In 4D, the only timepoint that falls within this interval is the 6h timepoint, and it does not look like this one is different from the control upon p38 inhibition. Thus I am left unconvinced by this data.

The reviewer’s comment can be broken into two separate concerns. One, why did we decide to measure cell size at time points that are longer than 20 hours post Torin-washout? Two, the referee is concerned by the fact that p38 inhibition did not block recovery of cell size in the early time points (6h).

To address the questions, we have now added detailed explanations in the Results section – Inhibition of p38 represses recovery of cell size, as quoted below:

“As a final test for the hypothesis that p38 plays a role in the maintenance of cell size, we asked if p38 activity is required for cells that are recovering from perturbations of cell size. […]This delay may hint that the cell growth cycle depends on a commitment point that is cell-cycle-stage-dependent, analogous to the restriction point that regulates the cell division cycle (Blagosklonny and Pardee, 2002).”

6) The logical flow of the paper is clear and compelling but with the one caveat other explanations might be found for the delayed kinetic response of p38 inactivation after mTOR inhibitor washout. For instance, it could simply be that the kinetics of p38 inactivation are slower than for mTOR recovery, i.e., that the small size is not directly the cause of p38 activation. In other contexts, how fast is p38 inactivated after a stimulus is removed? I don't see an obvious experiment to address this issue.

We agree with the reviewer’s critique. If the p38 pathway has slow inactivation kinetics, we should expect slow recovery of p38 activity even in cases where cell size is unperturbed. To examine this possibility, we performed two sets of new experiments which show that the sustained p38 activation after washout of mTOR inhibitor is due to changes in cell size rather than slow inactivation kinetics of the pathway. In the revised manuscript, we added several new figures of the results (Figure 5, Figure 5—figure supplement 3,Figure 5—figure supplement 5) and a detailed description in the main article text. For convenience of the referees, we add below a quote from the revised manuscript that addresses these new results:

“The sustained p38 activity, which proceeds hours after mTOR activity has resumed normal levels of activity, supports the hypothesis that it is the reduction in cell size and not inhibition of mTOR that activates p38. […] These results suggest that the sustained activity of p38 accompanying the recovery of cell size results in reduction in cell size rather than mTOR inhibition or slow inactivation kinetics of p38 signaling.”

7) The raw screen data does not appear to be reported in a supplementary table so it is not possible for the analysis to be confirmed, or for further information to be extracted from the data after publication. The raw data for each compound in the screen should be explicitly provided in a supplementary table so it is easy for readers to access.

We are happy to comply with this request. With the resubmission, we will provide two datasets of “raw data”.

Dataset #1: Images of cells from all wells in the drug screen (~300 GB).

Dataset # 2: Single-cell measurements that were obtained from image-processing of the raw images in dataset #1 (~ 6GB).

Due to the large size of the datasets, we will upload the data to a publicly-available platform.

8) The authors state that loss of p38 increases cell size variability, but little if any data is provided that directly support this important, central point.

We now added, to the revised manuscript, a figure (Figure 3—figure supplement 3) that shows the increased variance in p38-inhibited cells. We quote the reference to this new figure in the revised manuscript “… as expected from perturbations of size checkpoints, inhibition of p38 increases the variability in cell size (Figure 3—figure supplement 3).”

[Editors' note: further revisions were requested prior to acceptance, as described below.]

Reviewer #1:As before, this paper reports a detailed analysis of the relationship between cell size and cell cycle progression that includes a large screen for compounds that disrupt this relationship. The methods for studying this relationship are innovative, high throughput, and accurate, and the screen convincingly pulled out inhibitors of P38-MAPK as agents that specifically disrupt the normal proportional relationship between cell size and cell cycle stage. The revision is much improved. It is now more concise and easy to read, and contains many important details of methods that were previously missing. Some distracting, irrelevant data has been removed and a few additional tests and analyses have been added that enhance the presentation. The central points, namely that cell size and cell cycle stage are normally coupled in human cells, and that loss of P38 can sever this coupling, are very well supported. This function of P38 appears to be somewhat unique, as well as unexpected, and warrants a good report in the public literature. However the paper still could be more compelling if more were reported about how p38 senses normal size/cell proportionality, and how it affects the cell cycle to maintain size uniformity. With regards to this latter point, the authors suggest it may work via the cyclin kinase inhibitor p27, which show's levels changes consistent with a regulatory function. However the p27 expression data here is cursory, and it's function in size maintenance is never tested, which is unfortunate. I would like to encourage the authors to perform a few functional tests of P27, since demonstrating that it's response is meaningful could enhance the paper a lot.

To address this comment, we added in the new manuscript short discussions regarding the point. The discussion is quoted as below.

“By probing for the major regulators of G1-S transition, our results suggest that the cell cycle inhibitor p27 may play a role in how p38 affects cell cycle to maintain cell size uniformity (Figure 2). However, it is worth noting that, in addition to p27, previous literature has implicated several other G1 regulators as p38 effectors. It is, therefore, not unlikely that the influence of p38 on G1 length is redundantly mediated by multiple branches of signaling. In future investigations, it will be interesting to examine the specific mechanisms that explain how p38 promotes cell size uniformity and, crucially, identify how p38 ‘senses’ cell size.”